# *N*-glycan-modified α-L-iduronidase produced by transgenic silkworms ameliorates clinical signs in a Japanese macaque with mucopolysaccharidosis I

## Abstract

**Background** Mucopolysaccharidosis type I (MPS I) is an inherited lysosomal storage disorder (LSD) caused by recessive mutations in the α-L-iduronidase (*IDUA*) gene. Enzyme replacement therapy (ERT) utilizing terminal mannose-6-phosphate (M6P)-carrying *N*-glycans attached to therapeutic enzymes produced by mammalian cell lines has been clinically applied to several LSDs. Recent studies suggested an unidentified delivery pathway mediated by sialic acid-containing *N*-glycans. However, more economical platform development is required to produce large quantities of recombinant enzymes. Transgenic silkworms have been established as low-cost systems for expressing recombinant glycoproteins. Microbial endo-β-*N*-acetylglucosaminidases (ENGases) enable the transglycosylation of *N*-glycans to other types.

**Methods** We purified recombinant human IDUA from *IDUA* transgenic silkworm cocoons and performed ENGase-mediated transglycosylation. Furthermore, we performed intravenous enzyme replacement therapy in a Japanese macaque MPS I non-human primate model carrying a homozygous *IDUA* missense mutation.

**Results** Here we show the establishment of *IDUA* transgenic silkworms and purification of recombinant human IDUA from cocoons. As M6P- and sialic acid-containing *N*-glycans are not attached to purified hIDUA, we perform ENGase-mediated transglycosylation to obtain hIDUAs with M6P- and sialic acid-containing *N*-glycans (neoglyco-hIDUAs). Furthermore, we perform intravenous neoglyco-hIDUA replacement therapy in MPS I non-human primate model and succeed in improving the clinical signs and reducing the urinary glycosaminoglycan (GAG) levels.

**Conclusions** These glycotechnologies using transgenic silkworms and ENGases are expected to serve as platforms for developing therapeutic glycoproteins.

## Plain language summary

Lysosomal storage disorders (LSDs) are a group of inherited diseases related to metabolism. These can result in buildup of toxic material in the body due to defects in enzymes (a type of protein). One of the current treatments for LSDs is enzyme replacement therapy (ERT), in which functional human enzymes are given to patients. However, producing large quantities of these therapeutic enzymes can be challenging. Here, we describe a method to produce certain types of proteins using silkworms. We used this method to evaluate its therapeutic effects on a non-human primate model of one type of LSD and succeeded in improving the animal's symptoms. This method could be a promising approach for producing treatments for humans.

The lysosomal enzyme α-L-iduronidase (IDUA) hydrolyzes glycosaminoglycans (GAGs), including dermatan sulfate (DS) and heparan sulfate (HS)[1]. Mucopolysaccharidosis type I (MPS I) is an inherited lysosomal storage disorder caused by mutations in the *IDUA* gene and is classified into three forms according to severity: Hurler disease, Hurler/Scheie disease, and Scheie disease[2]. Loss of IDUA activity causes excessive intra- and extracellular accumulation of DS and HS, resulting in systemic clinical manifestations[3]. One of the current clinical therapeutic applications for lysosomal storage disorders is enzyme replacement therapy (ERT), in which recombinant human enzymes are administered intravenously to patients. In particular, mannose-6-phosphate (M6P)-type or high mannose-type glycans attached to the administered recombinant enzyme bind to cation-independent M6P receptors (CI-M6PR) or mannose receptors (MR) on the cell surface and are internalized by endocytosis[4–6]. In a recent study, bestronidase alfa, which is a clinically applied therapeutic enzyme for MPS VII, was enriched with sialic acid-containing *N*-glycans, resulting in its

✉e-mail: oishi.takao.5e@kyoto-u.ac.jp; kitoh@tokushima-u.ac.jp

distribution in the bones and heart of MPS VII mice after ERT[7], suggesting an unidentified delivery pathway mediated by sialic acid-containing *N*-glycans and an improved pharmacokinetics. Recombinant enzymes taken up into the cell via these pathways are transported to lysosomes and exert their therapeutic effects.

Human IDUA (hIDUA) is composed of 653 amino acid residues and has six intramolecular *N*-glycosylation sites, including M6P-containing glycans at Asn336 and Asn451[8], which are necessary for lysosomal transport. The high mannose-type glycan at Asn372 has been reported to bind to the catalytic pocket of hIDUA, which is essential for enzymatic activity[9]. A recombinant hIDUA with M6P-containing glycans[10] produced in mammalian cell lines, named laronidase, has been clinically applied as an ERT for MPS I[11]. However, developing a more economical platform for producing large quantities of recombinant lysosomal enzymes is necessary. In recent years, transgenic silkworms have been used as a low-cost and safe expression system for recombinant glycoproteins[12–17]. Although the insect cell expression system has a risk of antigenicity in humans due to the addition of α1,3 fucose-containing *N*-glycans[18], the recombinant glycoproteins produced in transgenic silkworm cocoons (TSCs) have few α1,3 fucose-containing *N*-glycans and show a similar glycan structure to those of mammals[12,19]. The recombinant human cathepsin A (hCTSA), a lysosomal enzyme purified from cocoons of transgenic silkworm expressing CTSA in the posterior silk glands, was shown to be enriched in high mannose-type *N*-glycan similar to that in mammals[20], however, it did not carry M6P- or sialic acid-containing glycans.

Recently, a glycotechnology was developed to transglycosylate the *N*-glycans attached to *N*-glycoproteins to other *N*-glycan-types using microbial endo-β-*N*-acetylglucosaminidases (ENGases)[21] and their mutants. Endo-M was found to hydrolyze the chitobiose structure (GlcNAcβ1-4GlcNAc) of *N*-glycans and to exert transglycosylation activity in the presence of *N*-glycopeptides, while the Endo-M N175Q mutant exhibited only transglycosylation activity[22]. Endo-CC was reported to exert both hydrolysis and transglycosylation activities, with the Endo-CC N180H mutant having higher transglycosylation activity with terminal sialic acid-containing biantennary glycans as donors[23]. ENGases have gained attention as a technology that can modify natural *N*-glycans into functional *N*-glycans.

In the present study, recombinant hIDUA with M6P- or sialic acid-containing *N*-glycans was prepared using a transgenic silkworm expression system and ENGase treatment. The generated recombinant hIDUAs with modified *N*-glycans (neoglyco-hIDUAs) were administered to a macaque with MPS I, identified non-human primate model by our team, to investigate their therapeutic effects.

## Methods
### Establishment of transgenic silkworm line overexpressing hIDUA in middle silk glands

The *piggyBac* vector pBac[Ser1UAS-*Bln*I, 3xP3-GFP] was generated by replacing TATA of the upstream activation sequence (UAS) sequence of the pBac[UAS-*Bln*I-UTR, 3xP3-GFP][24] with TATA of a *sericin*1 promoter. The *Eco*RI-digested fragment of [Ser1UAS-*Bln*I] was extracted from the pBac[Ser1UAS-BlnI, 3xP3-GFP]. The *Nhe*I-digested fragment of [A3KMO] was extracted from the pBac[A3KMO, UAS][24]. The pBac[Ser1p-GAL4VP16, 3xP3-DsRed2] was generated by replacing the actin3 promoter of the pBac[A3-GAL4VP16, 3xP3-DsRed2][25] with the *sericin*1 promoter. The *Asc*I-digested fragment of [Ser1p-GAL4VP16] was extracted from the pBac[Ser1p-GAL4VP16, 3xP3-DsRed2]. Three DNA fragments, the [Ser1UAS-*Bln*I], the [A3KMO], and the [Ser1p-GAL4VP16] were inserted into the pBacMCS[26], resulting the pBac[Ser1UAS-*Bln*I, Ser1p-GAL4VP16, A3KMO] was created.

To generate a transgenic line expressing hIDUA in its middle silk glands (T-12 line), the expression plasmid pBac[Ser1UAS-IDUA, Ser1p-GAL4VP16, A3KMO] was co-injected with the helper plasmid pHA3PIG into pre-blastoderm embryos of the *w-1 pnd strain*[27]. The hatched larvae were reared on an artificial mulberry diet at 25 °C. G1 embryos were screened under a stereo microscope, and a T-12 line was obtained.

### Establishment of transgenic silkworm line overexpressing hIDUA

The codons of hIDUA gene were optimized to suit the silkworm type, and an artificial IDUA gene was generated. To generate a transgenic silkworm line expressing hIDUA in middle and posterior silk glands (T-14 line), the optimized IDUA gene was inserted downstream of the UAS of the *piggyBac* vector pBac[SerUAS_Ser1intron_hr5/3xP3-EYFP_A3-Bla][14]. For the GAL4 strain that expresses the *GAL4* gene in the middle and posterior silk glands, the *Asc*I-*Spe*I fragment of the pBac[Ser1-GAL4/3xP3-DsRed][28] was inserted into the *Asc*I-*Bln*I site of the pBac[BmFibHL-GAL4/3xP3-DsRed][29], and the resultant plasmid was designated as pBac[Ser1-GAL4_-FibH-GAL4/3xP3-DsRed2]. The pBac[SerUAS_Ser1intron_hr5/3xP3-EYFP_A3-Bla] and pBac[Ser1-GAL4_FibH-GAL4/3xP3-DsRed2] were separately co-injected with the pHA3PIG into pre-blastoderm embryos of the *w-1 pnd* strain[27]. The hatched larvae were reared on an artificial mulberry diet at 25 °C. G1 embryos were screened under a fluorescence stereo microscope equipped with an EYFP or DsRed2 filer, and UAS-IDUA and Ser1-GAL4_FibH-GAL4 lines were obtained. Both lines were crossed to produce silkworms (T-14 line) harboring two transgenes, UAS-IDUA and Ser1-GAL4_FibH-GAL4. The cocoon shells were harvested six days after spinning.

### Extraction and purification of recombinant hIDUA

The *IDUA* TSCs were soaked in ultrapure water (5 mL per cocoon). After repeated decompression and atmospheric pressure, until they become translucent, the cocoons were shaken twice at 4 °C for 30 min. Subsequently, extraction buffer [20 mM sodium phosphate buffer (NaPB) containing 150 mM NaCl (pH 7.0)] was added (5 mL per cocoon). After repeated decompression and normal pressure for 5 min, the cocoons were shaken overnight at 4 °C. Samples were through a 0.22-μm-pore filter, and the extract and insoluble fractions were separated. The insoluble fraction was further shaken with newly added extraction buffer (5 mL per cocoon) for 4 h at 4 °C, and the extract was collected in the same tube. The extract was concentrated using an Amicon Ultra filter (30,000 MWCO; Merck Millipore, Burlington, MA). An equivalent amount of 20 mM NaPB (pH 7.0) supplemented with 2 M ammonium sulfate was added, and the mixture was incubated on ice for 30 min. The samples were then applied to a Hi Trap Butyl FF column (Cytiva, Marlborough, MA) equilibrated with 20 mM NaPB (pH 7.0) supplemented with 1 M ammonium sulfate. After washing, the bound proteins were eluted by decreasing concentrations of ammonium sulfate. The elution fraction (0.8–1.0 M ammonium sulfate) was dialyzed against 10 mM NaPB/0.5 M NaCl (pH 6.0) and then concentrated using the Amicon Ultra filter (30,000 MWCO). Purification with Phos-tag resin was performed as previously described[30] with modifications. Briefly, samples were applied to Phos-tag agarose (NARD Institute, Amagasaki, Japan) in a Poly-Prep Chromatography Column (Bio-Rad, Hercules, CA) balanced with 0.1 M Tris-acetate buffer (pH 7.5) supplemented with 20 mM zinc acetate. After washing the column with Tris-acetate buffer, bound proteins were eluted with Tris-acetate buffer supplemented with 1 M NaCl and 10 mM sodium phosphate. The eluate was dialyzed against 10 mM NaPB/0.5 M NaCl and concentrated using an Amicon Ultra filter (30,000 MWCO).

### SDS-PAGE analysis

Protein concentrations were measured using a *DC* protein assay (Bio-Rad) with BSA (Sigma-Aldrich, St. Louis, MO) as a protein standard.

For Coomassie brilliant blue (CBB) or silver staining, each sample (1 μg protein) was subjected to SDS-PAGE on a 10% (w/v) polyacrylamide gel. Subsequently, the gel was subjected to 0.02% (w/v) CBB R-350[31] (PhastGel Blue R-350, GE Healthcare Bio-Sciences AB, Uppsala, Sweden) or silver staining using a Dodeca Silver Stain Kit (Bio-Rad) according to manufacturer's protocol.

For blotting with Phos-tag biotin and *Sambucus sieboldiana* agglutinin (SSA) lectin, protein samples were separated on SDS-PAGE and electro-transferred onto Immobilon-P membranes (Merck Millipore) using the Trans-Blot SD Semi-Dry Transfer Cell (Bio-Rad). Membranes were blocked

with 50% (v/v) Blocking One/Tris-buffered saline at 25 °C for 1 h for SSA lectin, and probed with 1 mM Phos-tag BTL-111 (NARD)/10 mM zinc nitrate hexahydrate/HRP-streptavidin (Sigma-Aldrich) at 25 °C for 1 h, or 2 µg mL⁻¹ biotin labeled SSA lectin (MGC Woodchem, Tokyo, Japan) overnight at 4 °C. The bound probes were visualized by treatment with an anti-biotin HRP-linked antibody (1:1000 dilution with blocking buffer, 7075, Cell Signaling Technology, Beverly, MA) at 25 °C for 1 h, followed by chemiluminescence detection using Western Lightning Ultra (Perkin Elmer, Waltham, MA) in a ChemiDoc XRS+ system (Bio-Rad).

## Enzyme assays

Samples were diluted with ultrapure water if needed. Then IDUA activities toward 4-methylumbelliferyl-α-L-idopyranosiduronic acid 2-sulfate disodium (4-MU-IdoA, 2 mM; Biosynth, St. Gallen, Switzerland) were measured in 0.1 M sodium acetate buffer (pH 4.5) supplemented with 500 mM NaCl[32]. Briefly, 15 µL of 4-MU-IdoA and the sample were mixed, incubated at 37 °C for 30 min, and the reaction was terminated with 380 µL of 0.2 M Glycine-NaOH (pH 10.7). Then, 300 µL of each was applied to a 96-well plate, and fluorescence was measured (Ex. 355 nm and Em. 460 nm).

## ENGase treatment

For Endo-D treatment, HM-IDUA was incubated with Endo-D (100 units per 1 mg IDUA protein, New England Biolabs, Ipswich, MA) in 50 mM NaPB (pH 7.5) supplemented with 500 mM NaCl for 24 h at 37 °C. To remove Endo-D, 0.1 mL of chitin-resin (New England Biolabs) equilibrated with 50 mM NaPB (pH 7.5)/0.5 M NaCl was added per 1 mg protein of IDUA, and rotated at 4 °C overnight. After adding 15 mL of 50 mM NaPB (pH 7.5)/0.5 M NaCl, the sample was applied to a poly-prep chromatography column, and the flow-through fraction was dialyzed against 10 mM NaPB/0.5 M NaCl (pH 6.0) and then concentrated using an Amicon Ultra filter (30,000 MWCO). The sample was labeled as GlcNAc-IDUA.

For Endo-M (N175Q) treatment, GlcNAc-IDUA was incubated with Endo-M (N175Q) (1 mU per 1 mmol GlcNAc-IDUA protein, Tokyo Chemical Industry, Tokyo, Japan) and Man$_5$(6P$_2$)GlcNAc$_2$-methoxyphenyl (1 mol per 1 mmol GlcNAc-IDUA protein, Tokyo Chemical Industry) in 50 mM MES buffer (pH 6.0) for 24 h at 30 °C. To remove the unreacted Man$_5$(6P$_2$)GlcNAc$_2$-methoxyphenyl, the sample was dialyzed against Tris-buffered saline using an Amicon Ultra filter (30,000 MWCO). The sample was labeled as M6P-IDUA.

For Endo-CC (N180H) treatment, HM-IDUA was incubated with Endo-CC (N180H) (1 mU per 1 nmol IDUA protein, Fushimi Pharmaceutical, Marugame, Japan) and α2,6-sialylglycopeptide (1 µmol per 1 nmol IDUA protein, Fushimi Pharmaceutical, CAS No. 189035-43-6) in 50 mM NaPB (pH 7.5) for 24 h at 30 °C. To remove Endo-CC (N180H) and unreacted α2,6-sialylglycopeptide, the sample was purified using a Hi Trap Butyl FF column in the same manner as TSCs, dialyzed against 20 mM NaPB/150 mM NaCl (pH 6.0), and concentrated using an Amicon Ultra filter (30,000 MWCO). The sample was labeled as SG-IDUA.

## Site-specific *N*-glycosylation analysis

The *N*-glycan profile of each glycosylation site in recombinant hIDUAs purified from the TSCs (T-12 line) was determined using LC-MS/MS. Briefly, an aliquot of protein sample of approximately 60 µg was denatured and reduced in 50 µL of 7 M guanidine chloride supplemented with 0.25 M Tris-HCl (pH 8.0), 5 mM EDTA, and 10 mM DTT at 37 °C for 90 min, and then alkylated with 24 mM iodoacetic acid at 25 °C for 30 min in the dark. The sample was quenched with 4 mM DTT, desalted using a PD-MiniTrap G-25 (Cytiva), and lyophilized. Carboxymethylated protein was digested using sequence grade trypsin or chymotrypsin (Promega, Madison, WI) at 1:50 enzyme:sample ratio in 50 µL of 25 mM Tris-HCl (pH 7.5) at 37 °C for 6 h. Digestion was stopped by adding 1 µL of 10% (v/v) formic acid. The digests were analyzed on an Ekspert nanoLC 400 system with the 3 µm C18 NANO HPLC CAPILLARY COLUMN75-3-15 with a NANOSpray III source connected to TripleTOF 6600 mass spectrometer (Sciex, Tokyo, Japan) in the positive ion mode. Mobile phase A was 0.1% (v/v) formic acid,

and mobile phase B was 0.1% (v/v) formic acid in acetonitrile. The flow rate was set at 300 nL per min. Digests were eluted by a linear gradient from 5–55% B for 50 min after 1 min initial flow at 5% B. The *m/z* range of MS was set at 700–2000 for glycopeptide analysis and 400–2000 for confirmation of amino acid sequence. The top 5 most abundant precursor ions were selected for MS/MS. Glycopeptides were assigned based on the presence of signature ions, such as *m/z* 204 (HexNAc) and 366 (Hex-HexNAc), and ions of peptide + HexNAc in the MS/MS spectrum or by mass differences between glycan units and other glycopeptides[33]. The *N*-glycan structure was deduced based on the molecular mass of the carbohydrate moiety. The glycan profiles at Asn110, 190, 336, 372, and 415 were obtained through trypsin digestion, whereas the profile at Asn451 was obtained through chymotrypsin digestion. The intensities of glycopeptides were calculated using the LC-MS Peptide Reconstruct software tool (with peak findings). For confirmation of amino acid sequence, MS data were analyzed using BioPharmaView software 3.0 (Sciex) against human IDUA sequence (UniProt: P35475 IDUA_HUMAN) with 2 missed cleavage, a mass tolerance of 10 ppm. The unidentified amino acid sequence was confirmed by MS data from Asn-N digest and Lys-C digest.

## Cell culture

Primary skin fibroblasts were isolated from auricular specimens of an MPS I Japanese macaque. Skin specimens were minced into 2–3 mm pieces with scissors, placed on 0.1% gelatin-coated culture plates, and cultured in Dulbecco's modified Eagle's medium with high glucose (FUJIFILM Wako, Osaka, Japan) supplemented with 15% FBS, 1× nonessential amino acids (FUJIFILM Wako), 1× GlutaMAX supplement (Gibco, Waltham, MA, USA), 1 mM sodium pyruvate (FUJIFILM Wako), 55 µM 2-mercaptoethanol (Gibco), and 100 U mL⁻¹ penicillin and 100 µg mL⁻¹ streptomycin (FUJIFILM Wako) at 37 °C with 5% CO$_2$. The cells were expanded by serial passages with 0.25% trypsin/EDTA. Primary skin fibroblasts of a healthy Japanese macaque were isolated in the previous study[34].

## Enzyme replacement assay for MPS I fibroblasts

Each type of fibroblast was seeded onto a collagen-type I-coated 35-mm dish (AGC Techno Glass, Haibara, Japan). HM-, M6P-, and SG-IDUA (1.0 µg protein per mL medium in each experiment) were added, followed by incubation for 24 h. The cells were sonicated in 50 mM sodium acetate buffer/150 mM NaCl containing 1% (v/v) nonidet P-40 (NP-40) (Roche, Mannheim, Germany)/protease inhibitors (1 µM pepstatin A, 2 mM EDTA, and 20 µM leupeptin). Then, the sample was centrifuged at 18,000 × g for 5 min, and the supernatant was collected as a cell extract. In some experiments, 5 mM M6P (Sigma-Aldrich) was added 30 min before enzyme addition.

The IDUA delivery imaging was performed as previously described[30] with modifications. Briefly, HM-, M6P-, and SG-IDUA (30 µg protein in 0.1 M sodium bicarbonate buffer, pH 8.3) were mixed with 10 mM Acidi-Fluor ORANGE-NHS (AFO, Goryo Chemical, Sapporo, Japan) at 25 °C for 2 h. The resultant AFO-labeled IDUAs were dialyzed against 0.1 M sodium bicarbonate buffer (pH 8.3) with an Amicon Ultra filter (10000 MWCO). AFO-labeled IDUAs (each 1 µg protein) were added to MPS I fibroblasts, followed by incubation for 24 h. The cells were viewed with a BIOREVO BZ-9000 device (Keyence, Osaka, Japan).

## Animals

All procedures used in this study followed the Guidelines for the Care and Use of Non-human Primates, provided by Primate Research Institute, Kyoto University (PRI)[35] after approval by the Institutional Animal Welfare and Care Committee (2015–157, 2016–114, 2017–113, 2018–017, 2019–050, 2020–029, 2021–132, 2022–058, 2023–142 and 2024–074).

Three MPS I macaques were born and housed as members of Wakasa group, a naturalistic group consisting of approximately 60 individuals in an open enclosure (858 m²) for several years. Then, the MPS I macaques were transferred to an indoor hospital individual/paired

cages (D650 mm × W780 mm × H800 mm or D760 mm × W900 mm × H850 mm, for pair or a small group, two to four cages are connected together). Macaques were fed on 100 g pellets (AS, Oriental Yeast, Tokyo, Japan) twice a day and vegetables three times a week, with occasional special treats of apples, bananas, peanuts, and dried bananas. Environmental enrichment, such as various feeders, wooden toys, climbing structures, and swings, was provided depending on the housing condition. For indoor cages, the room temperature was kept at 20–27 °C. Concerning outdoor open enclosures, atmospheric temperatures range from −2 to 37 °C throughout the year.

An MPS I macaque (#2) was used in ERT study as described below, and other two macaques (#1 and #3) were not. The macaques were anesthetized with a combination of ketamine hydrochloride (50 mg mL$^{-1}$ Ketalar, 5 mg kg$^{-1}$; Daiichi Sankyo Propharma, Tokyo, Japan), medetomidine hydrochloride (0.025 mg kg$^{-1}$ Dorubene; Meiji Seika Pharma, Tokyo, Japan) and midazolam (0.125 mg kg$^{-1}$, Sand Co., Ltd., Tokyo, Japan) followed by intravenous injection of secobarbital sodium (10–20 mg kg$^{-1}$, Ional sodium; Nichi-Iko Pharmaceutical Co., Ltd., Toyama, Japan), and euthanized by exsanguination followed by phosphate-buffered saline (PBS) perfusion at the age of 8 (#1 and #2) and 5 (#3) under the guidelines for the Care and Use of Non-human Primates[35], as they showed deterioration of general condition. The interval between the completion of ERT and euthanasia in #2 was 2 years.

### DNA sequence

Genomic DNA was obtained from blood samples. The sequences of *IDUA* gene were determined using Ex Taq hot start version (Takara Bio, Kusatsu, Japan). The forward and reverse primers were as follows: 5′-TGGCC GCCTCAGTACCAC-3′ and 5′-CTGAACTACTACGATGCCTGC-3′. PCR amplification was performed using the following conditions: initial denaturation at 94 °C for 10 min, 30 cycles of denaturation at 98 °C for 10 s, annealing at 65 °C for 30 s, and extension at 72 °C for 15 s, followed by a final extension at 72 °C for 10 min.

### Structural prediction

Pathogenicity due to variants at H262 residue in human IDUA was predicted by AlphaMissense[36] in TogoVar database[37].

### Hematological and blood chemistry analyses

Complete blood count and blood chemistry analyses of samples from d1 to d282 were performed using pocH-100iV Diff (Sysmex Corporation, Kobe, Japan) and FUJIFILM DRI-CHEM 7000 V (FUJIFILM, Tokyo, Japan), respectively. Samples from d379 to d421 were analyzed by Animal Medical Technology (Nagoya, Japan).

### Enzyme replacement therapy and clinical evaluation

HM-IDUA (first course), M6P-IDUA (second course), and SG-IDUA (third course) from the T-14 line were administered intravenously (0.58 mg per kg body weight) to an MPS I macaque. A 64-d and 173-d rest period, respectively, was allowed between courses. The macaque was anesthetized every other week six times for enzyme administration, followed by once for evaluation for the first and second courses. For the third course, three times for enzyme administration followed by three times for evaluations. The anesthesia was induced with a combination of ketamine hydrochloride (50 mg mL$^{-1}$ Ketalar, 5 mg kg$^{-1}$), medetomidine hydrochloride (0.025 mg kg$^{-1}$ Dorubene) and midazolam (0.125 mg kg$^{-1}$), and general anesthesia was maintained with 1.5–3% sevoflurane (Sevofulo; Zoetis Japan, Tokyo, Japan) with 100% O$_2$ as carrier gas administered through a facial mask. Radiography, blood and urine sampling, observation of gingiva and laryngeal region, and whole-body photography were performed before enzyme administration to evaluate the effects of the previous ERT. An intravenous catheter was placed in a saphenous vein, and Ringer acetate was administered intravenously for 5 min. Then, the enzyme was infused via a catheter at 1 mL h$^{-1}$ for 5 min, thereafter the infusion rate was doubled every 5 min up to 32 mL h$^{-1}$ and then maintained 32 mL h$^{-1}$ for 90 min. During the infusion, measurement of skin thickness (cheek, femoral

region, and lateral abdomen), head and body length, and ultrasound examination of heart, abdomen, and knee joint were performed. Physical activity and resting and sleeping time were evaluated using Plus Cycle[38] (Japan Animal Referral Medical Center, Kawasaki, Japan) during and after the second and third courses. Plus Cycle was placed inside a crafted nylon collar[39] and covered with adhesive wrap bandage.

### Pathological analysis

Tissue samples were excised from the major organs during necropsy and fixed in 10% neutral-buffered formalin. Samples were processed routinely, and paraffin-embedded tissue sections were stained with hematoxylin and eosin (H&E) for histopathological examination.

### Measurement of GAGs in urine

Urine samples were collected from macaques and frozen at –80 °C. To determine urinary GAG content, 100 μL of urine sample (no dilution for WT macaque, 100-fold dilution with ultrapure water for MPS I macaque) was measured using the Blyscan Sulfated Glycosaminoglycan Assay kit (Biocolor, Carrickfergus, UK) according to manufacturer's protocol. Urinary creatinine levels were measured using a Urinary Creatinine Assay Kit (Cell Biolabs, San Diego, CA) or Fuji Dri-Chem CRE-P III (FUJIFILM Medical, Tokyo, Japan) according to the manufacturer's protocol. Chondroitin 4-sulfate and creatinine were used as standards.

### Quantification of GAGs using high-performance liquid chromatography (HPLC)

Urine GAGs were purified as previously described[40]. Briefly, urine samples were digested in 50 mM Tris-acetate buffer (pH 8.0) supplemented with 0.33% actinase E (Kaken Pharmaceutical, Tokyo, Japan) at 45 °C for 3 h. Next, 15 mM acetic acid supplemented with 10% NaCl was added to the sample solution. The mixture was heated at 94 °C for 5 min, cooled on ice, and centrifuged at 2300 × $g$ for 15 min. The supernatant was transferred to an Amicon Ultra-0.5 mL Centrifugal Filter (Merck Millipore) pretreated with 20 μL of 0.1 M NaOH. After centrifugation at 2300 × $g$ for 20 min, the retained GAG fraction was replaced by PBS.

Purified GAGs were digested with chondroitinase (Chase) ABC, Chase AC-II, Chase B, heparitinase (HSase), and heparinase (Hepase) at 37 °C for 4 h. Chase ABC from *Proteus vulgaris* (EC 4.2.2.20 & 4.2.2.21), Chase AC-II from *Arthrobacter aurescens* (EC 4.2.2.5), Chase B (EC 4.2.2.19), HSase from *Flavobacterium heparinum* (EC 4.2.2.8), and Hepase from *Flavobacterium heparinum* (EC 4.2.2.7) were purchased from Seikagaku Corporation (Tokyo, Japan). Digested samples were labeled with 2-aminobenzamide, as previously described[41] with modification. Briefly, an aliquot (5 μL) of a derivatization reagent mixture (0.25 M 2-aminobenzamide/1 M NaCNBH$_3$/30% (v/v) acetic acid in dimethyl sulfoxide) was added to the disaccharide sample, and the mixture was incubated at 65 °C for 2 h. After excess 2-aminobenzamide was removed by chloroform extraction, the derivatized disaccharide was analyzed using HPLC on an amine-bound silica PA-G column (4.6 × 250 mm; YMC, Kyoto, Japan) with a linear gradient of NaH$_2$PO$_4$ at a flow rate of 1 mL min$^{-1}$ at 25 °C as previously reported[42]. Eluates were monitored using an RF-10AXL fluorometric detector (Shimadzu, Kyoto, Japan) at excitation and emission wavelengths of 330 and 420 nm, respectively.

### Detection of anti-hIDUA antibodies in plasma

Plasma samples were collected from macaques and frozen at –80 °C. HM-IDUA was diluted to 10 ng/μL of bicarbonate buffer (pH 9.6) and 100 μL was applied to each well of a 96-well high-adsorption plate (Sumitomo Bakelite, Tokyo, Japan), and incubated overnight at 4 °C in the dark. The IDUA solution was removed, and each well was blocked with 200 μL of blocking buffer (0.5% BSA/PBS) at 25 °C for 1 h. After washing with 200 μL of PBS, 100 μL of plasma sample (1:100 dilution with blocking buffer) was applied and incubated at 25 °C for 2 h. After washing, 100 μL of goat anti-monkey IgG H&L HRP-conjugated (1:1000 dilution with blocking buffer; ab112767, Abcam, Cambridge, UK) was applied and incubated at 25 °C for

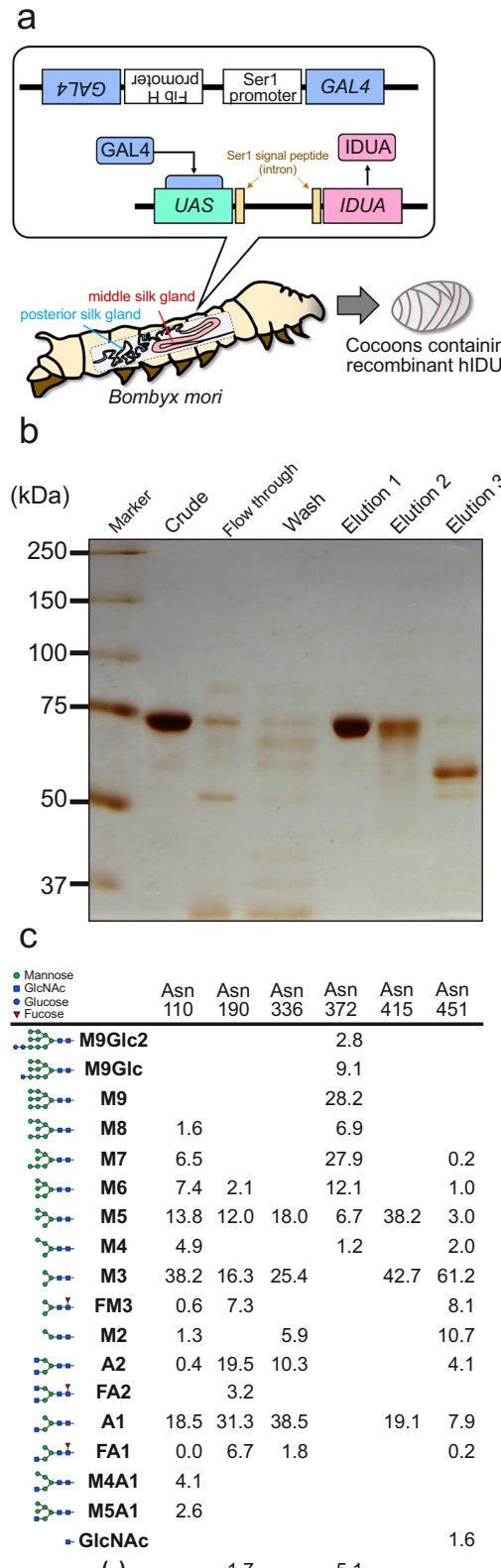

**Fig. 1 | Extraction, purification, and *N*-glycan profiles of hIDUA proteins from TSCs. a** Schematic illustration of the middle and posterior silk gland-specific IDUA expression of the transgenic silkworm (*Bombyx mori*) using the GAL4-UAS system (T-14 line). Ser1: Sericin1, Fib H: Fibroin heavy chain. **b** IDUA proteins were purified using butyl column chromatography. Each fraction was separated using SDS-PAGE, followed by silver staining. Each lane contained 1 µg of protein. Elution fraction 1, 2, and 3 contains 0.8–1.0 M, 0.65–0.8 M, and 0–0.65 M ammonium sulfate, respectively. Each experiment was repeated at least twice. **c** *N*-glycan compositions and percentage abundances. Glucose (blue circle), GlcNAc (blue square), mannose (green circle), and fucose (red triangle). M9Glc2 Glc₂Man₉GlcNAc₂, M9Glc GlcMan₉GlcNAc₂, M9 Man₉GlcNAc₂, M8 Man₈GlcNAc₂, M7 Man₇GlcNAc₂, M6 Man₆GlcNAc₂, M5 Man₅GlcNAc₂, M4 Man₄GlcNAc₂, M3 Man₃GlcNAc₂, FM3 Man₃GlcNAc₂Fuc, M2 Man₂GlcNAc₂, A2 GlcNAc₂Man₃GlcNAc₂, FA2 GlcNAc₂Man₃GlcNAc₂Fuc, A1 GlcNAcMan₃GlcNAc₂, FA1 GlcNAcMan₃GlcNAc₂Fuc, M4A1 GlcNAcMan₄GlcNAc₂, M5A1 GlcNAcMan₅GlcNAc₂.

reproduced due to limited number of MPS I macaques. Instead, we tracked changes in the macaque over time.

## Results

### Establishment of *IDUA* transgenic silkworm and purification of recombinant hIDUA from cocoons

To obtain *N*-glycan-containing recombinant hIDUA, we established transgenic silkworms expressing hIDUA in the middle and posterior silk glands (Fig. 1a). We extracted proteins from the *IDUA* TSCs and subjected them to hydrophobic interaction chromatography (Fig. 1b). The hIDUA protein was identified as a single band at ~75 kDa in the elution 1 fraction. Consequently, we observed that the specific activity of purified hIDUA (872 µmol h⁻¹ per mg protein) was increased by 1.9-fold compared with that of crude (459 µmol h⁻¹ per mg protein). Based on the total IDUA activity, we estimated the recovery rate to be 85.6%, suggesting that ~85–100 µg of purified IDUA could be obtained from one cocoon of T-14 line. Compared with the T-12 line (expressing in the middle silk glands), allowing for ~20 µg of purified IDUA per cocoon, the higher yield of the T-14 line was suggested to reflect higher hIDUA expression. We then used LC-MS/MS to determine the *N*-glycan profile of each glycosylation site (Fig. 1c). We found that the major *N*-glycan structures attached to purified hIDUA were M3, A1, M5, A2, M6, M7, and M9. Most major *N*-glycans were in high- and pauci-mannose forms such as M7 and M3. Notably, we seldom detected fucose-containing *N*-glycans. Unfortunately, purified IDUA did not contain any M6P-type and sialic acid-containing *N*-glycans consistent with previous analyses of *N*-glycan-containing hCTSA[20]. Furthermore, LC-MS/MS analysis confirmed that the amino acid sequence of the purified hIDUA was the same as that of the reported human IDUA.

### Modification of *N*-glycans to M6P-containing *N*-glycans in purified hIDUA

To replace the *N*-glycans of purified hIDUA with M6P-containing *N*-glycans, we treated the purified hIDUA with Endo-D and Endo-M (N175Q) in a stepwise manner (Fig. 2a). To this end, we use biantennary M6P-containing *N*-glycan derivatives as donors. We observed that the molecular weight of purified hIDUA (high mannose-IDUA; HM-IDUA) was reduced after Endo-D treatment (GlcNAc-IDUA), whereas it was increased again after Endo-M (N175Q) treatment (M6P-IDUA), as indicated by the SDS-PAGE followed by CBB staining (Fig. 2b). *N*-glycan profiles of GlcNAc-IDUA indicated that Endo-D treatment markedly reduced the M3, M5, A2, and A1 levels at Asn110, 190, 336, and 451 (Supplementary Fig. 1). However, we found that the M3–M5-enriched *N*-glycans at Asn415 were barely cleaved by Endo-D, suggesting that the catalytic site of Endo-D was not accessible to these glycans. To confirm the presence of M6P-containing *N*-glycans, we performed Phos-tag biotin blotting. Accordingly, we detected a broad band at 80–85 kDa in the Endo-M-treated fraction, suggesting the coexistence of unreacted GlcNAc-IDUA and M6P-IDUA with one to four M6P-containing *N*-glycans (Fig. 2c). To remove unreacted GlcNAc-IDUA, we purified the Endo-M-treated fraction using Phos-tag agarose and

90 min. A peroxidase assay kit for ELISA (Sumitomo Bakelite) was used to measure the absorbance at 450 and 540 nm.

### Statistics and reproducibility

Sample size and reproducibility are described in Legend. No statistical analysis was performed in this study. In vivo experiments could not be

**Fig. 2 | Modification of M6P-containing *N*-glycans in purified hIDUA. a** Schematic illustration of preparing M6P-IDUA from high mannose-IDUA (HM-IDUA). MP, methoxyphenyl. **b, c** HM-IDUA was treated with Endo-D and Endo-M (N175Q). Each sample was separated using SDS-PAGE, followed by CBB staining (**b**) and Phos-tag biotin blotting (**c**). Each lane contained 1 µg of protein. **d, e** Purification of M6P-IDUA from a mixture of GlcNAc- and M6P-IDUA using Phos-tag agarose. Each fraction was separated using SDS-PAGE, and followed by CBB staining (**d**) and Phos-tag biotin blotting (**e**). Each lane contained 1 µg of protein. All experiments were repeated at least twice.

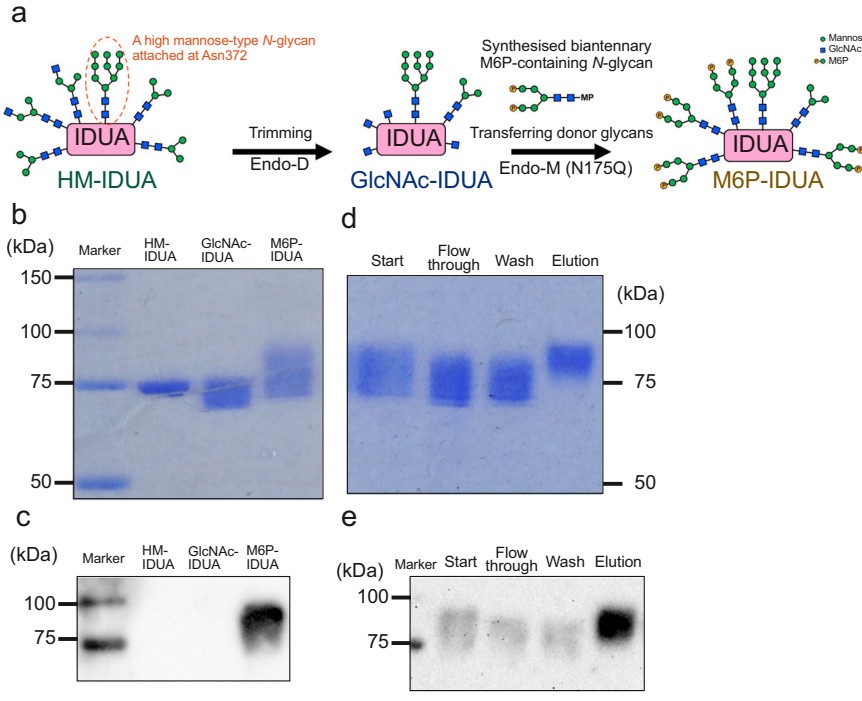

showed the presence of low-molecular-weight IDUA in the flow-through and wash fractions, whereas high-molecular-weight IDUA was observed in the elution fraction (Fig. 2d). Furthermore, blotting with Phos-tagged biotin showed a clear signal of approximately 80 kDa in the elution fraction (Fig. 2e). We obtained approximately 4.4 mg of M6P-IDUA from approximately 25.9 mg of HM-IDUA, with a recovery rate of 16.9%. These results indicated that ENGases enabled the preparation of M6P-type glycan-containing hIDUA.

## Modification of *N*-glycans to sialic acid-containing *N*-glycans in purified hIDUA

Next, we investigated the modification of *N*-glycans of HM-IDUA to sialic acid-containing glycans using Endo-CC (N180H) (Fig. 3a). We treated HM-IDUA with Endo-CC (N180H) to cleave high/pauci-mannose-type *N*-glycans and transglycosylate them to α2,6-sialylglycan (SG), followed by purification using a butyl column. Both SDS-PAGE followed by CBB staining and blotting using SSA that recognizes α2,6 sialic acid, indicated the acquisition of α2,6-SG-containing hIDUA (SG-IDUA) (Fig. 3b, c). We obtained approximately 6.9 mg of SG-IDUA from ~15.0 mg of HM-IDUA, with a recovery rate of 45.9%.

## Identification of spontaneous MPS I macaques

To date, *Idua* knockout mice[43] and domestic species, including cats and dogs with mutations in the orthologous *IDUA* have been reported as MPS I animal models[44,45]. Although these animal models show a pathophysiology similar to that of patients with MPS I and have been used to study the mechanisms of pathogenesis and treatment strategies, studies in non-human primate models with biological and behavioral characteristics similar to those of humans are also needed. Recently, we found individuals with MPS-like clinical signs in a group of Japanese macaques (*Macaca fuscata*), which originated from Wakasa, Tottori, Japan, and had been introduced to the PRI in the early 1970s and have been kept separately in their original groups to conserve their genetic background. Compared with wild-type (WT), these MPS-like macaques exhibited a short nasal bridge and wrinkles under the eyes, and redness around the eyes shortly after birth, resembling the gargoyle-like facial features characteristic of patients with MPS I (Fig. 4a). Although no corneal opacity was observed in these macaques, they exhibited clinical

findings, including developmental retardation (short stature and slow teething), protruding tongue, gingival hyperplasia, thickened skin, kyphosis, and knee joint contracture (Fig. 4b, c). Vertebral deformities and mitral valve regurgitation were also identified (Fig. 4d, e). The first male case of MPS-like macaque (#1) walked slowly and had difficulty getting food in the group, and eventually, the macaque became difficult to live independently in the group and was hospitalized. There were no signs of hydrocephalus in the MPS-like macaque (#1). In the other two macaques (#2, female and #3, male), there was mild expansion of lateral ventricles, although the degree was as that sometimes found in healthy Japanese macaques. We found no abnormality in taking food pellets from the box outside the cage or in their social relationship with cage-mate macaques or caretakers. Since there were no obvious central nervous system clinical signs, including psychomotor retardation, these macaques are considered to be an attenuated type of MPS I (Scheie disease or Hurler/Scheie disease). As expected, we detected a missense mutation based on a single-nucleotide variant (c.786 C > A) in the *IDUA* allele of the identified Japanese macaques (Fig. 4f). In addition, no recessive mutations were found in the other 20 genes responsible for lysosomal disease (Supplementary Table 1). The mutation was well correlated with MPS-like clinical signs, and we found a recessive inheritance only in Wakasa group of Japanese macaques (Fig. 4g). Although the H262 mutation has not been reported in patients with MPS I, the amino acid sequences of human and macaque IDUA are 96% conserved (Fig. 4h), and we predicted the H262Y and H262R variants in humans to be likely pathogenic using AlphaMissense[36] tool. In addition, as H262 in hIDUA has been reported to be positioned near the substrate binding site[9], it was suggested that mutant IDUA (p. H262Q) has reduced affinity for substrates. We detected that the urinary GAG levels, including heparin, HS, chondroitin sulfate (CS), and DS, were higher in the MPS I macaques than in WT (Fig. 4i, Supplementary Tables 2 and 3). These physical characteristics, gene mutations, and urinary GAG accumulation indicated that the identified macaque had MPS I as verified by the loss of IDUA activity. The lifespan of a normal Japanese macaque is 20–30 years, whereas the MPS I macaques showed progressive deterioration of general conditions including appetite loss, diarrhea, development of pressure ulcers, severe edema, and eventually circulatory failure at the age of 8 (#1 and #2) and 5 (#3) years, and the

**Fig. 3 | Modification of sialic acid-containing *N*-glycans in purified hIDUA. a** Schematic illustration of preparing SG-IDUA from HM-IDUA. **b, c** HM-IDUA was treated with Endo-CC (N180H). Each sample was separated using SDS-PAGE, followed by CBB staining (**b**) and SSA lectin blotting (**c**). Each lane contained 1 μg of protein. All experiments were repeated at least twice.

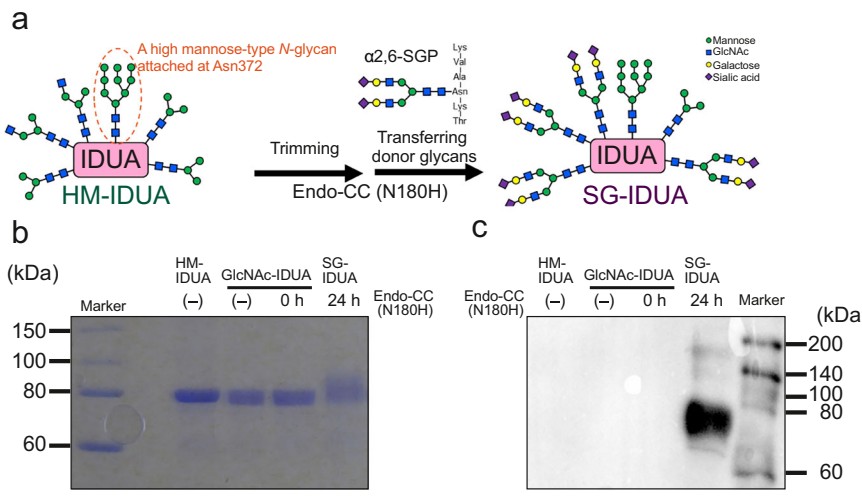

macaques were euthanized under the guidelines for the Care and Use of Non-human Primates[35]. At necropsy, we observed thickened soft palate, liver nodules, and deformed heart valves. Pathological studies in the gingiva and mitral valve showed deposits of foamy macrophages characteristic of MPS (Fig. 4j).

### Effect of enzyme replacement with hIDUAs on MPS I macaque-derived fibroblasts

We then evaluated the effect of enzyme replacement with M6P- and SG-IDUA on MPS I macaque auricular skin-derived fibroblasts. We found that M6P-IDUA restored IDUA activity via CI-M6PR (Supplementary Fig. 2a). In contrast, treatment with SG-IDUA showed CI-M6PR-independent recovery. To visualize the delivery of M6P- and SG-IDUA to lysosomes in living cells, we labeled each enzyme with AcidiFluor ORANGE (AFO). While almost no signal was observed with AFO-labeled HM-IDUA, signals correlating with enzyme replacement effects were detected in cells treated with AFO-labeled M6P- and SG-IDUA, indicating that the endocytosed M6P- and SG-IDUA were delivered to lysosomes (Supplementary Fig. 2b).

### Effect of enzyme replacement with hIDUAs on an MPS I macaque

We analyzed the effect of enzyme replacement with HM-, M6P-, and SG-IDUA in an MPS I macaque (6-year-old female, #2 in Fig. 4g). Considering the dose of laronidase (0.58 mg per kg body weight, weekly) and the high sensitivity of macaques to general anesthesia, we intravenously administered HM-IDUA (0.58 mg per kg body weight, every other week, six times) in the first course, followed by M6P-IDUA (0.58 mg per kg body weight, every other week, six times) in the second course and SG-IDUA (0.58 mg per kg body weight, every other week, three times) in the third course (Fig. 5a). To evaluate three enzymes using the same individuals, we allowed a 64-d and 173-d rest period, respectively, between courses. Two weeks after the last administration of HM-IDUA, we observed the exposure of the embedded canines and deepening of the gingival sulcus, which aggravated again 2 years after completion of enzyme administration (Fig. 5b). We also evaluated clinical changes, including increased daily activities during and after the second and third courses (Fig. 5c) and decreased puncture resistance suggesting recovery of skin elasticity as early as on d14. We detected that the urinary total GAG and heparin/HS levels were slightly decreased in the first course and even more decreased in the second and third courses (Fig. 5d, e), and this suppressive effect was maintained until 4 weeks after the final administration of SG-IDUA. At the beginning of the third course, urinary heparin/HS levels remained low (Fig. 5e), even though total urinary GAG levels were elevated again (Fig. 5d). We found CS/DS levels increased at d378 and DS level decreased at d434 (Supplementary Fig. 3a), suggesting that M6P-IDUA is highly distributed in the heparin/HS-accumulating

tissues. In addition, GAG levels increased again 2 years after ERT completion (Supplementary Fig. 3b).

Previous studies have shown that ERT with laronidase produces anti-IDUA IgG in more than 90% of patients with MPS I, some of which are neutralizing antibodies that inhibit the uptake of the administered enzyme into the target tissue[46]. To evaluate the safety of TSC-derived hIDUA, we measured the anti-IDUA antibody titers in plasma samples from the macaque. We accordingly observed the production of anti-IDUA antibodies during the first course of treatment (Fig. 5f). Likewise, M6P- and SG-IDUA also induced the production of anti-IDUA antibodies. However, we observed that even after antibody production, clinical signs, such as decreased urinary GAG levels and increased skin elasticity, were still present, suggesting that these antibodies had little neutralizing activity. Compared to healthy Japanese macaques[47], the MPS I macaque had elevated liver enzymes, including alanine aminotransferase and gamma-glutamyl transpeptidase following ERT (Supplementary Table 4), and was given oral medication, including 0.5 g taurine, 0.2 g ursodeoxycholic acid, and 0.1 g diisopropylamine dichloroacetate, from d149 to d275. Importantly, we did not observe any serious adverse reactions, including anaphylactic shock, throughout the study. These results demonstrated the efficacy and safety of M6P- and SG-IDUA from TSCs for ERT in an MPS I macaque.

## Discussion

In this study, we transglycosylated *N*-glycans attached to recombinant hIDUA purified from TSCs to M6P- or SG-type using ENGases and evaluated their therapeutic efficacy in a spontaneous MPS I macaque model (Fig. 6).

Recently, transgenic silkworms have emerged as a GMP-standard production platform for glycoproteins, including antibody drugs[48,49], due to their ability to produce large amounts of glycoproteins with complex macromolecular structures at a lower cost than mammalian cell expression systems and their suitability for scale-up and industrialization[15,49]. Therefore, we used this platform as a source of *N*-glycosylated hIDUA for ERT. In this study, we extracted and purified a large amount of recombinant hIDUA without M6P- or sialic acid-containing *N*-glycans from TSCs (Fig. 1). This may be due to differences in maturation and localization of the M6P-modifying enzymes, between mammalian and insect cells[50]. Mannose receptors are highly expressed in macrophages and rarely in mammalian skin fibroblasts. Although there are some reports of other types of recombinant enzymes that are taken up by fibroblasts via the mannose receptor[51–53], HM-IDUA could hardly be incorporated into macaque fibroblasts. To deliver TSC-derived hIDUA to a broad peripheral tissue, we first modified

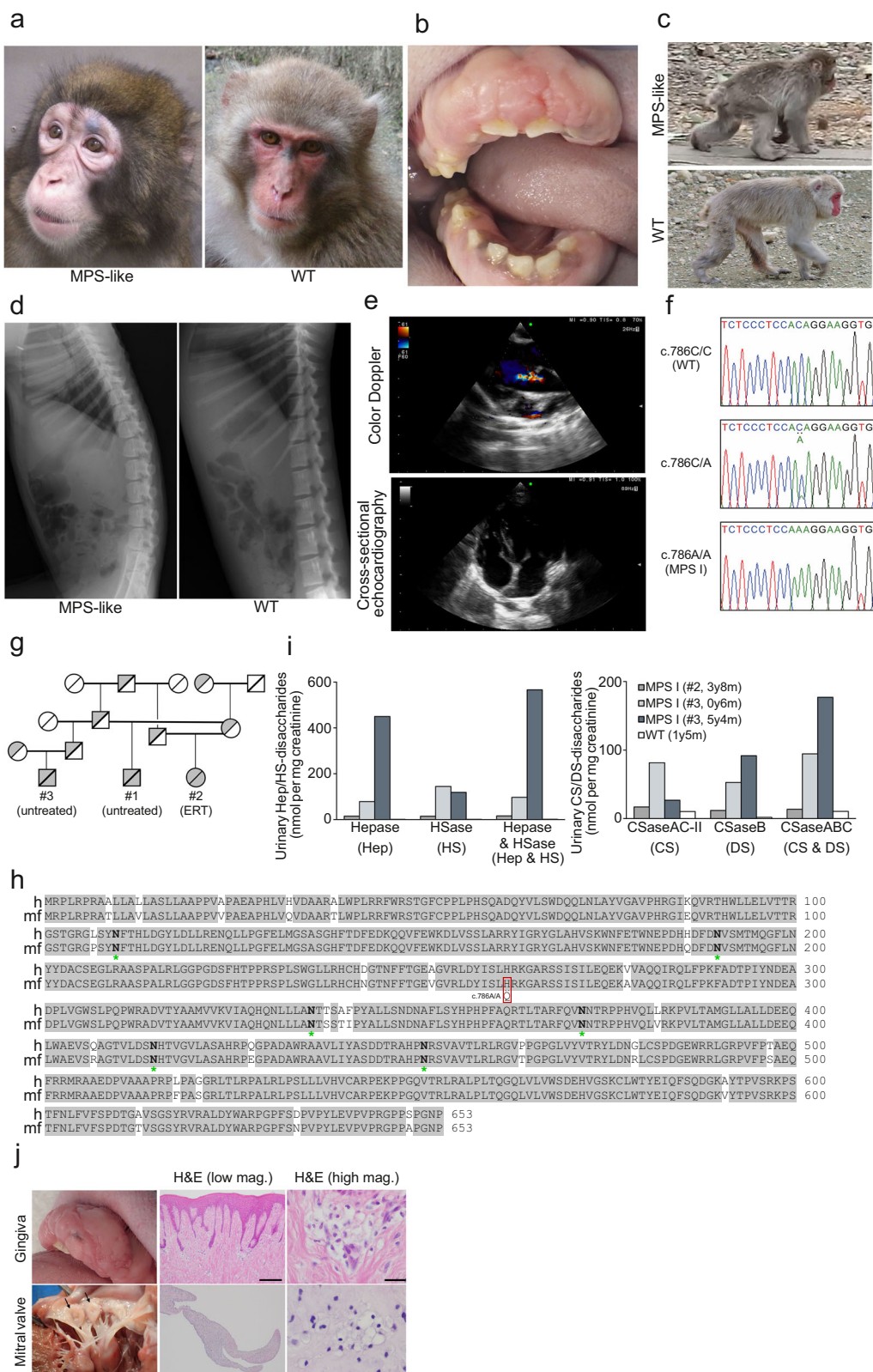

recombinant hIDUA with biantennary M6P-containing *N*-glycans using ENGases (Fig. 2). The HM-IDUA was enriched in M3–M5 pouch/high mannose-type *N*-glycans (Fig. 1c) and was selectively cleaved by Endo-D, maintaining the high mannose-type *N*-glycans attached to Asn372, which are essential for IDUA activity. Because Endo-M (N175Q) is specialized for transglycosylation activity[22], we used it for

the attachment of M6P-containing *N*-glycans. We expected that at least one and up to four biantennary M6P-containing *N*-glycans would be attached to M6P-IDUA, with the M6P content ranging between 2 and 8, higher than that of laronidase by approximately 2.5[10]. Mammalian M6P-containing *N*-glycans have a phosphate group added to the terminal mannose; accordingly, in this study, a phosphate group was added to the

**Fig. 4 | Identification of spontaneous MPS I macaques. a** Facial appearance of MPS-like (male, 4 years old, #1) and WT (female, 4 years old) macaques. **b** Protruding tongue and gingival hyperplasia in an MPS-like macaque (female, 5-year-old, #2). **c** Walking posture of MPS-like (male, 4 years old, #1) and WT (female, 10 years old) macaque. The posture reflects the contracture of the knee joint and kyphosis. **d** X-ray images showing the skeleton of MPS-like (male, 4 years and 4 months old, #1) and WT (male, 3 years and 11 months old) macaques. The spine shows kyphosis, and the ventral side of the vertebrae of the lumbar spine is rounder than that in WT macaques. **e** Color Doppler image (female, 5 years old, #2) showing aortic regurgitation and four-lumen cross-sectional echographic image (female, 6 years and 1 months old, #2) showing thickening of the mitral valve in a female MPS-like macaque. **f** Determination of mutations in the *IDUA* gene of MPS I macaques. Sequences representing WT (c.786 C/C), heterozygous (c.786 C/A), and homozygous (c.786 A/A) *IDUA* mutation. **g** Family tree of the Japanese macaque in this study. Circles and squares indicate female and male, respectively. White and gray indicate WT and mutant, respectively. **h** Amino-acid sequences of human (h) and macaque (mf) IDUA. The sequences are aligned using the Smith-Waterman algorithm. Shaded boxes indicate identity in two polypeptides. The asterisk denotes a *N*-glycosylation sequon. **i** Quantification of urinary GAG levels in WT and MPS I macaques. Hep: heparin, HS: heparan sulfate, CS: chondroitin sulfate, DS: dermatan sulfate. MPS I (female, 3 years 8 months old, #2), MPS I (male, 6 months and 5 years 4 months old, #3), WT (female, 1 year 5 months old). **j** Gross and microscopic appearances of gingival and valvular lesions in an MPS I macaque (female, #2). Deformation of the mitral valve (arrows) resulted in regurgitation. Histopathologically, foamy macrophages are observed in the collagenous fibrous tissue of the gingiva (top) and are distributed over the entire region of the mitral valve (bottom). H&E staining. Bars are 500 µm (middle) and 25 µm (right), respectively.

terminal mannose of the synthesized biantennary M5. M6P-IDUA showed cellular uptake (Supplementary Fig. 2a), indicating that biantennary M6P-containing *N*-glycans are useful for ERT via CI-M6PR. Next, to investigate cellular uptake via sialic acid-containing *N*-glycans, we prepared SG-IDUA by modifying the *N*-glycans of HM-IDUA with α2,6-SG (Fig. 3). As Endo-CC (N180H) has high transglycosylation activity with terminal sialic acid-containing biantennary *N*-glycans as donors[23], we used it to modify *N*-glycans of HM-IDUA into sialic acid-containing *N*-glycans. The addition of *N*-glycans or enrichment of sialic acid-containing *N*-glycans has been reported to improve the pharmacokinetic properties of therapeutic proteins, including erythropoietin and ENPP1, by increasing their steric bulk and avoiding receptor-mediated clearance[54]. SG-IDUA was successfully taken up by cells in an MR- and CI-M6PR-independent manner, suggesting that sialic acid-containing *N*-glycans promote cellular uptake. However, the mechanism of cellular uptake of SG-IDUA remains unclear, and further studies using human cell lines, including kidney and neuronal cells, are needed to elucidate the target receptors, such as selectins and Siglecs[55]. Alternatively, it is possible that sialic acid-containing *N*-glycans improved pharmacokinetics and enhanced uptake via MR. Future studies in mice are needed to investigate the pharmacokinetics of SG-IDUA. The combinations of transgenic silkworms and chemoenzymatic methods using ENGases are promising approaches for producing designer glycoproteins (neoglycoenzymes) with functional *N*-glycans, whereas immunoblotting and SDS-PAGE of M6P- and SG-IDUA showed smear bands in the elution fraction (Figs. 2 and 3), indicating the presence of different types of *N*-glycosylation in the proteins. Further studies using model proteins with only one *N*-glycan, such as GM2A[56], are needed to consider the potential implications of non-homogenous proteins on efficacy, safety, and productivity. Also, additional modification and purification steps are required to prepare the transglycosylated enzymes. Transgenic silkworms could be useful as an expression system of recombinant proteins, the expressed proteins could show instability which may be caused by differences in glycosylation patterns[15], it is necessary to address issues related to scale-up and cost performance improvement in the future.

Patients with Hurler disease, a severe form of MPS I, have central nervous system symptoms and develop severe multiorgan failure, including bone deformities and cardiomyopathy[57], whereas patients with moderate or mild MPS I, Hurler/Scheie disease or Scheie disease do not exhibit central nervous system symptoms[58]. MPS I macaques shared clinical signs with patients with MPS I, such as developmental retardation, kyphosis, vertebral deformities, mitral valve regurgitation, and high urinary GAG level (Fig. 4). As MPS I macaques showed no obvious signs in the morphology of the brain and almost no difference in daily food-taking in the cage or relationships with cagemate macaques or caretakers, these macaques can be the model not for Hurler disease but for moderate or mild MPS I, Hurler/Scheie disease or Scheie disease. Following hIDUA administration, the macaque became more active in daily life, possibly due to improved peripheral tissue clinical signs, including increased skin elasticity and suppressed inflammatory

responses. Urinary GAG levels are known to be increased in patients with MPS I and are thus used as a diagnostic and therapeutic parameter[59,60]. As the source of urinary GAGs are serum GAGs filtered in the kidneys or GAGs in the connective tissue of the kidneys and surface layer of the urothelium[61], the HM-IDUA-dependent decrease in urinary GAG levels may be the result of their incorporation via MR into reticuloendothelial cells. Alternatively, because DS and HS are also present on the cell surface[62], it is possible that HM-IDUA binds to the cell surface DS/HS, and the complex is taken up into the cell by endocytosis. Treatment with M6P-IDUA reduced the urinary GAG levels compared with those after HM-IDUA administration (Fig. 5c), suggesting that M6P-IDUA was taken up by peripheral tissues, including renal tissue and reticuloendothelial cells in a CI-M6PR-dependent manner. SG-IDUA also reduced the urinary GAG levels (Fig. 5c), suggesting that SG-IDUA were incorporated into peripheral tissues via binding to cell surface DS/HS or unidentified sialic acid receptors. Urinary heparin/HS levels remained low at the beginning of the third course (Fig. 5e), the ERT effect of SG-IDUA could be partially due to the tolerance of M6P-IDUA. Since the enormous cost, years, and effort required to use the MPS I macaque model, it will be necessary to combine MPS I model mice, WT non-human primate models, and MPS I macaques to evaluate the efficacy and safety of other modalities such as gene therapy in the future studies.

In ERT, intravenously administered recombinant enzymes are rarely distributed to tissues with low blood flow, including the cardiac valves, bones, and joints. This is a common problem with ERT, hematopoietic stem cell transplantation[63], and gene therapy[64]; thus, the identification of unique pathways to these tissues is needed. Recently, bestronidase alpha, which is enriched in sialic acid-containing *N*-glycans while maintaining M6P-type glycans, was reported to be distributed in the bones and hearts of MPS VII mice[7]. Therefore, we hypothesized that the recombinant enzyme could be incorporated in a sialic acid-containing *N*-glycan-dependent manner. Unfortunately, although the administration of hIDUAs to the MPS I macaque improved some clinical signs, none of these hIDUAs showed any efficacy against valvular clinical signs or bone deformities, suggesting little distribution of hIDUAs to bones or heart valves. Hence developing therapeutic enzymes that can be distributed to these tissues by utilizing this combination of transgenic silkworms and ENGases, which can modify not only natural-type glycans but also artificial glycans, is warranted in the future.

In conclusion, we prepared recombinant hIDUAs with M6P- and sialic acid-containing neoglyco-hIDUAs using TSCs and ENGases. Neoglyco-hIDUAs were demonstrated to exhibit therapeutic effects, including the improvement of clinical signs and reduction in urinary GAG levels in the MPS I macaque. Technologies for producing glycoproteins from TSCs and modifying *N*-glycans using ENGases are expected to serve as platforms for developing targeted therapeutic glycoproteins.

## Data availability
The uncropped gel and membrane images are shown in Supplementary Figs. 4–6. The source data for graphs and charts are presented in Supplementary Data 1. DNA sequence data obtained in this study were deposited

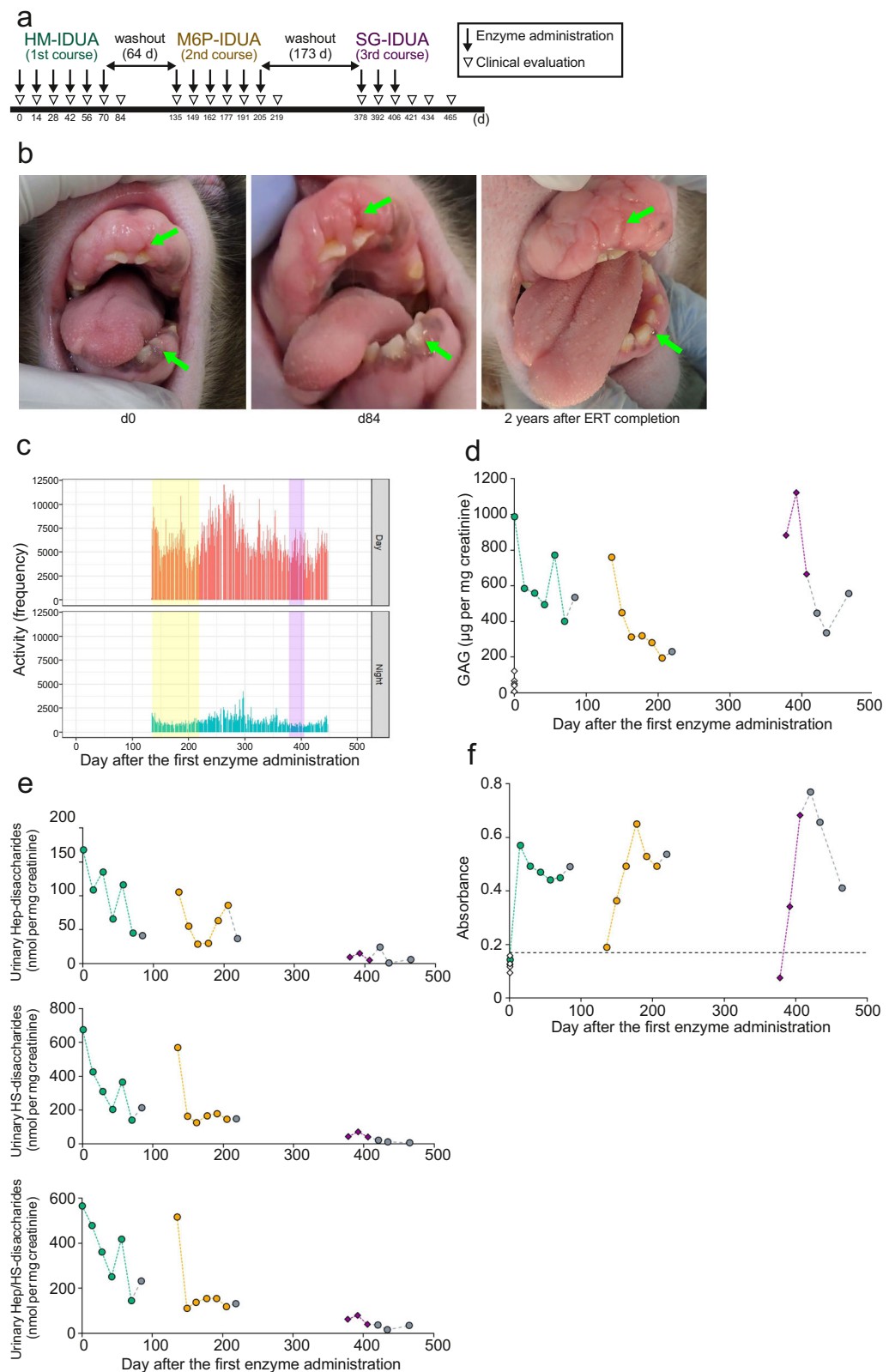

**Fig. 5 | Effect of enzyme replacement with hIDUAs on an MPS I macaque.**
**a** Experimental design of IDUA administration to an MPS I macaque (female, 6-year-old, #2 in Fig. 4g). **b** Gingival hyperplasia in an MPS I macaque before (left, d0), after (middle, d84), and 2 years after completion (right) of enzyme administration. Arrows indicate canines and gingival sulcus. **c** Activity level measured using Plus Cycle activity monitor. The activity level appeared to transiently increase following the second course ERT. The yellow and pink bands indicate the second and the third course, respectively. **d** Quantification of urinary GAG levels. Green, yellow, and purple indicate values after HM-, M6P-, and SG-IDUA administration, respectively. Gray circles indicate values measured without enzyme administration. White squares indicate WT macaques (female, $n = 5$). **e** Quantification of urinary Hep and HS levels in the MPS I macaque. **f** Evaluation of anti-IDUA antibody levels. Green, yellow, and purple indicate values after HM-, M6P-, and SG-IDUA administration, respectively. White squares indicate WT macaques (female, $n = 5$). The dashed line indicates cutoff value (mean $+2\,SD$ of WT).

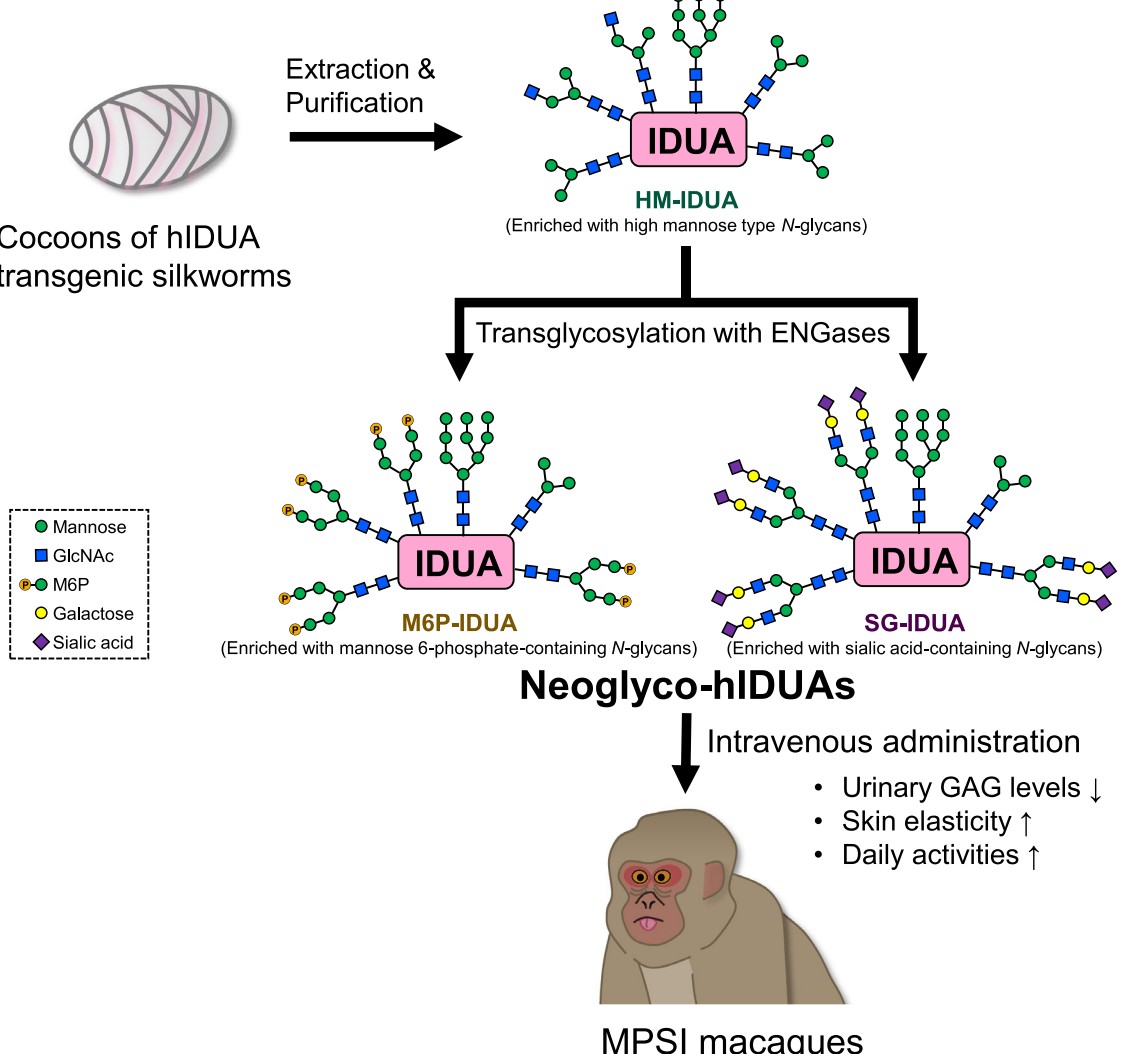

**Fig. 6 | A proposed model for ERT using neoglyco-hIDUAs.** *N*-glycans-containing recombinant hIDUA purified from TSCs are transglycosylated to M6P- or SG-type using ENGases, and administrated to a spontaneous MPS I macaque model.

in NCBI-Genbank (https://www.ncbi.nlm.nih.gov/genbank/) with the following accession numbers: PV389050 (MPS I #1), PV389051 (MPS I #2), PV389052 (MPS I #3), and PV389053 (WT). All other data provided in this article and supplementary files are available from the corresponding author upon request.

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

## Acknowledgements
The authors would like to thank Dr. Toshiki Tamura, Dr. Hiroaki Machii (National Institute of Agrobiological Sciences), and Mari Kondo (The University of Tokyo) for the establishment of the transgenic silkworm line; Dr. Nana Kawasaki (Yokohama City University) for *N*-glycan profiles; Dr. Kenji Yamamoto (Ishikawa Prefectural University), Dr. Toshiyuki Inazu, Dr. Katsuji Haneda (Tokai University), Yuki Kubo (Masuda Chemical Industries), Minoru Suda, and Masahito Dozaki (Fushimi Pharmaceutical) for ENGases and *N*-glycan donors; Dr. Makoto Yamamoto (Japan Animal Referral Medical Center) for providing Plus Cycle raw data; Mayuko Oe-Ike (Tokushima University) for secretarial assistance; Mayumi Morimoto and Miyuki Ido (Kyoto University) for technical assistance; Tokyo Chemical Industry for providing Endo-M (N175Q) and $Man_5(6P_2)GlcNAc_2$-methoxyphenyl. The authors thank Editage (www.editage.com) for the English language editing. This work was supported by the Japan Society for the Promotion of Science KAKENHI (grant numbers 26293120 and 17H04102 to K.I. and 18K06442 to T.O.), the Ministry of Agriculture, Forestry and Fisheries of Japan (Research Project for Sericultural Revolution & Agriculture, Forestry and Fisheries Research Council) to H.S. and K.I., Japan Agency for Medical Research and Development (AMED, grant number JP18ak0101074) to A.I.-W., AMED TR-SPRINT Seed A (grant number A-187) to K.I., Japan Science and Technology Agency (A-STEP & grant number JPMJTM17DB) to K.I., the Mizutani Foundation for Glycoscience to K.I., and Fushimi Pharmaceutical Co. Ltd. to K.I., the grant for NHP-A (Non-human Primates-Associates) to T.O., and the Cooperative Research Program of Primate Research Institute, Kyoto University. This work was also supported by Ryosuke Shibato and 71 crowdfunding backers (Takeyuki Akiyama, Chiyoko Ogawara, Takashi Ota, Kanzaki Seiichi, Takashi Sekine, Hiroyuki Takahashi, Yuriko Yamada, Japan MPS Research Association, Japan MPS Patient and Family Group, and others) on the "academist" platform (https://academist-cf.com/projects/31) to K.I. The funders had no role in the study design, data collection, decision to publish, or manuscript preparation.

## Author contributions
Conceptualization, K.I.; data curation, K. Kitakaze; investigation, C.S., K. Kitakaze, Y. Sasai, S. Nishioka, I.K., M.S., K.T., T.I., A. Harazono, A.M., A.K., M.I., T.M.-N., Y.G., A. Hirata, Y.T., T.M., K. Kiriyama, J.T., and S. Nadanaka; writing–original draft, C.S., K. Kitakaze, and Y. Sasai; writing–review & editing, I.K., K.T., A. Harazono, A.M., A.K., M.I., T.M.-N., Y.G., A. Hirata, S. Nadanaka, A.I.-W., T.K., H.K., Y. Suzuki, T.O., H.S., and K.I.; visualization, K. Kitakaze, T.M.-N., and T.O.; project administration; T.O. and K.I.; funding acquisition, T.O., H.S., and K.I. All authors approved the manuscript for publication.

## Competing interests
The authors declare the following competing interests: K.I. received funding from Fushimi Pharmaceutical Co. Ltd. A.M. and T.K. are employees of Fushimi Pharmaceutical Co. Ltd. All other authors declare no conflicts of interest.

## Additional information

Chika Shinoda[1,15], Keisuke Kitakaze [ID][2,15], Yuya Sasai[1], So-Ichiro Nishioka[1], Isao Kobayashi[3], Megumi Sumitani [ID][3], Ken-Ichiro Tatematsu[3], Tetsuya Iizuka[3], Akira Harazono[4], Ai Mitani[5], Akihisa Kaneko[6,13], Masanori Imamura[6,14], Takako Miyabe-Nishiwaki[6,13], Yasuhiro Go [ID][7,8], Akihiro Hirata[9], Yoshie Takeuchi[10], Teru Mizuno[1], Kei Kiriyama[1], Jun Tsukimoto[1], Satomi Nadanaka [ID][11], Akiko Ishii-Watabe[4], Takashi Kinoshita [ID][5], Hiroshi Kitagawa [ID][11], Yasuyuki Suzuki[12], Takao Oishi [ID][6,13,16] ✉, Hideki Sezutsu[3] & Kohji Itoh [ID][1,10,16] ✉

[1]Department of Medicinal Biotechnology, Graduate School of Pharmaceutical Sciences, Tokushima University, Tokushima, Japan. [2]Department of Pharmacology, Kawasaki Medical School, Kurashiki, Okayama, Japan. [3]Institute of Agrobiological Sciences, National Agriculture and Food Research Organization, Tsukuba, Ibaraki, Japan. [4]Division of Biological Chemistry and Biologicals, National Institute of Health Sciences, Kawasaki, Kanagawa, Japan. [5]Carbohydrate Research & Development Department, Fushimi Pharmaceutical Co., Ltd. 307 Minato-machi, Marugame, Kagawa, Japan. [6]Primate Research Institute, Kyoto University (PRI), Inuyama, Aichi, Japan. [7]Graduate School of Information Science, University of Hyogo, Hyogo, Japan. [8]The Exploratory Research Center on Life and Living Systems, National Institutes of Natural Sciences, Okazaki, Aichi, Japan. [9]Joint Department of Veterinary Medicine, Faculty of Applied Biological Sciences, Gifu University, Gifu, Japan. [10]Department of Medicinal Biotechnology, Faculty of Pharmaceutical Sciences, Tokushima University, Tokushima, Japan. [11]Laboratory of Biochemistry, Kobe Pharmaceutical University, Kobe, Hyogo, Japan. [12]Medical Education Development Center, Gifu University, Gifu, Japan. [13]Present address: Center for the Evolutionary Origins of Human Behavior (EHUB), Kyoto University, Inuyama, Aichi, Japan. [14]Present address: Department of Medical Neuroscience, Graduate School of Medical Sciences, Kanazawa University, Ishikawa, Japan. [15]These authors contributed equally: Chika Shinoda, Keisuke Kitakaze. [16]These authors jointly supervised this work: Takao Oishi, Kohji Itoh. ✉e-mail: oishi.takao.5e@kyoto-u.ac.jp; kitoh@tokushima-u.ac.jp

