## [Transparent Peer Review file · Communications Medicine]

***N*-glycan-modified α -L-iduronidase produced by transgenic silkworms ameliorates clinical signs in a Japanese macaque with mucopolysaccharidosis I**

Corresponding Author: Professor Kohji Itoh

Version 0:

Reviewer comments:

Reviewer #1

(Remarks to the Author)

This manuscript is a well written and describing paper which is focused in assessed a new and low cost expression system to obtain a human recombinant IDUA, with appropriate glycosylations obtained through a transgenic silkworms and a transglycosylation method. Likewise, once the rhIDUA was purified and characterized, they evaluated the therapeutic effect in a non-human primate.

This work has novelty, involving the use of a not commonly used expression system for the production of recombinant proteins and a new method to obtain the desired glycosylation pattern. And the use of MPSI-like macaque to evaluate the effect of the enzyme it's very interesting.

However, in the discussion, the authors assure that the use of the transglycosylation method and silkworm as an expression system allows to obtain a higher homogeneity in the glycosylation pattern. This is only a partial truth, since the immunoblot and SDS-PAGE show a reduction in the smeared band in the elution lane, however is still smeared which indicate the presence of different types of glycosylations in the protein. So maybe the affirmation should be improve the heterogeneity, but not produce homogeneous glycosylations proteins.

In Figure S2, where the authors show the effect of IDUA treatment in MPSI macaque cells, it is not entirely clear the significant difference between untreated and treated cells. The bar error in M6P-IDUA untreated is high, so in the end the data obtained is almost equal to the treated cells. Similar results are observed in HM-IDUA and SG-IDUA between treated and non-treated cells. Authors present results of duplicate and triplicate data in the same analysis, which is questionable.

I'm wondering about the results shown in the enzyme delivery images because it has been reported that recombinant proteins can be internalized through mannose receptors, so at least some of the administered HM protein should be taken up. The image of SG-IDUA shows a blue background suggesting overexposure in the microscope. There seems to be no homogeneity between the images.

So, if the HM-IDUA did not show internalization in fibroblasts, why the authors use this protein in the macaque? what was the purpose of that? Why this protein works in vivo and not in vitro?

So, in terms of productivity, if this is a promiscuous protein for further ERT to MPSI, is this expression system a good platform to scale up the production of the recombinant enzyme? is the transglycosylation process and efficient method to improve the glycosylation pattern of the protein?

In terms of macaque evaluation, the use of only one animal (understanding that having an animal model of these diseases is complicated, and bearing in mind that this is a spontaneous model) is a too preliminary data to ensure that this is an effective therapy. So how do we know that if we use the same animal to evaluate three similar proteins, the therapeutic result of the second and/or third protein used is not due to the tolerance developed by the animal because it has previously been injected with other similar proteins, and not inherent to the therapeutic being evaluated?

In the figure 5e Why the urinary GAGs in the third administration start in almost zero and maintain in this low level? why the behaviour is different to the first and second administration where the initial point is higher than the post evaluating days? Between first and second protein the accumulation is recovered, but between second and third maintain low? there is no discussion about that in the manuscript

Reviewer #2

(Remarks to the Author)

In this manuscript, the authors present a novel strategy for producing N-glycosylated recombinant IDUA. Currently, most of the lysosomal enzymes for ERT, including the one approved for MPS I, are produced in mammalian cells, resulting in high prices and low yields. Therefore, it is crucial to develop novel strategies for the production of ERTs at lower prices, which may increase access to therapies. In this context, this manuscript represents a valuable contribution to this field. The description of the novel animal model is also an important contribution to the MPS and LSD fields. Nevertheless, there are some limitations to the impact of the results, and in some cases, the conclusions are not fully supported by the results.

1. Figure 2 illustrates that the treatment with Endo D and Endo M produces a broader band than that of HM-IDUA. This suggests the presence of a non-homogeneous protein, which contrasts with the authors' assertion that this approach may produce homogeneous N-glycans in comparison to other platforms. It is essential to quantify and discuss the homogeneity of the N-glycans present in M6P- and SG-IDUA. Furthermore, it is necessary to consider the potential implications of this non-homogenous proteins on efficacy, safety, and productivity.
2. It is recommended that the kinetic evaluation of the three recombinant enzymes produced in the manuscript be conducted.
3. The absence of N-glycan profiles for both 6P- and SG-IDUA precludes the drawing of certain conclusions. It is recommended that these profiles be included. With regard to SG-IDUA, the authors posit that cell uptake is mediated by the presence of sialic acid. However, they do not rule out the possibility of mannose receptor-mediated cell uptake, given that it has been observed in high mannose content lysosomal enzymes. The authors must exclude the possibility that the presence of sialic acid increases the half-life of the protein, as this has been widely reported and may facilitate uptake through mannose receptors.
4. The authors noted that SG-IDA exhibited reduced activity relative to M6P-IDUA. However, no statistically significant difference was observed between the two proteins in terms of their activity.
5. If Iaronidase is administered on a weekly basis, it would be beneficial to understand the rationale behind the administration of the new enzymes every other week.
6. The in vivo evaluation was conducted on a single animal that received the three treatments over a period of approximately 15 months. As observed in MPS I patients, it is crucial to consider the natural history of the disease when analyzing these results. Therefore, it is recommended to include the follow-up of a non-treated MPS I animal of the same age and during the same evaluation period. This would permit a more comprehensive analysis of the potential impact of the therapy on GAG levels, skin elasticity, exposure of canines, or daily activity, which are the endpoints highlighted by the authors.
7. Figure 5C. The authors indicate that there is an increase in activity following the infusion of M6P- and SG-IDUA. However, there is a lack of data regarding HM-IDUA. Additionally, there is an activity increase during the resting period between M6P- and SG-IDUA infusions. This increase merits further discussion.
8. The authors discuss about the impact on skin elasticity as well as the lack of impact in bone and cardiac disease. Nevertheless, this results are not presented in the manuscript.
9. The evidence from in vivo studies is inconclusive regarding the superiority of M6P- or SG-IDUA compared to HM-IDUA. It is uncertain whether M6P- or SG-IDUA will be cost-effective, given the need for additional modification and purification steps, which would increase the cost of the therapy.

Reviewer #3

(Remarks to the Author)

Lysosomal storage disorders are well-known targets for enzyme replacement therapy. In the presented study mucopolysaccharidosis type I, caused by defects in the L-iduronidase (IDUA) gene, is treated with a novel recombinant and glycoengineered IDUA. The authors used transgenic silkworms for the production of high protein amounts and improved its

ability for enzyme replacement therapy by chemoenzymatic modifications of IDUA N-glycans. These glycoproteins were applied to enzyme replacement therapy of mucopolysaccharidosis in an ape model. In summary, it is a comprehensive work, joining glycobiochemistry and glycoengineering with successful use in a promising therapeutic method, suited for medical applications in the future.

There is one major point which should be addressed in a revised version of the manuscript. The authors showed quantitative N-glycan analysis of original recombinant IDUA and of pretreated GlcNAc-IDUA. However, a detailed N-glycan analysis of the completely modified glycoproteins, M6P-IDUA and SG-IDUA, is missed. This is of particular importance, as these proteins are administered in enzyme replacement therapy, and its structure and composition should be known as exact as possible. Phos-tag affinity chromatography or lectin blotting are possible indicators of successful modification of N-glycans. Nevertheless, the amounts of N-glycans carrying mannose 6-phosphate and sialic acid, respectively, which might be influenced by incomplete modification or subsequent degradation, could not be drawn from the experiments.

Further points:

1. Page 6, chapter "Establishment of IDUA transgenic silkworm and purification of recombinant hIDUA from cocoons" and following figures. Description of N-glycan structures: Give a complete list of abbreviations used for N-glycan structures once in the manuscript. In addition, omit "-PA" from the structures. Use " α 1,3" and " α 2,6" without hyphen. Finally, analysis of amino acid sequence is not described in the Methods part.
2. Page 7. Avoid "as indicated by the Coomassie brilliant blue (CBB) staining". The method is SDS-PAGE; CBB is only a part of it, which must not be indicated in the main text when described in the methods. Please check the whole manuscript for respective modifications.
3. Page 6/7, "Modification of N-glycans...". Give masses of IDUA used for starting the experiments and their mass yields after the final purification step. This is important to estimate the scale necessary to reveal sufficient amounts of glycoproteins for enzyme replacement therapy.
4. Methods, general. Intention of the movement of a part of methods to supporting information is unclear for the reviewer. All methods should be joined in one chapter. Furthermore, re-check all chapters of the Methods part for exact description of the methods. Some necessary corrections are listed below, but the reviewer cannot not guarantee to get all gaps. It must be ensured, that all experiments can be completely repeated by other scientists.
5. Page 15, "After washing, the bound proteins were eluted by increasing the NaPB concentration." is wrong. The proteins were eluted by decreasing concentrations of ammonium sulfate.
6. Page 15, "CBB and silver staining". Change to "SDS-PAGE analysis" (see 2.). Furthermore, give a reference for CBB staining; there are hundreds of different protocols published.
7. Page 16, "Enzyme assays". Describe the assays in more detail. Ref. 48 is not sufficient here. The activity was measured for purified enzymes and in cell lysates, which requires different protocols.
8. Page 16, "ENGase treatment". Use "GlcNAc" instead of GN for N-acetylglucosamine. Define "MP". Give the exact structure for the α 2,6-sialylglycopeptide. Is it the same structure as shown in Fig. 3? Finally, describe purification method ("was applied to a Hi Trap Butyl FF column") in detail, or refer to the method described in page 15, if it is identical.
9. Page 17, "Blotting with...". Give blocking buffer and its incubation time. Describe "visualized using an anti-biotin HRP-linked antibody" in more detail (exact antibody, concentration, incubation time).
10. Page 17, "Site-specific N-glycosylation analysis". "...desalted using a PD- MiniTrap G-25"; give elution buffer. "0.1% formic acid"; indicate w/v or v/v. "Peptides were analyzed using a TripleTOF 6600 mass spectrometer (Sciex)"; describe method in detail or give a reference.
11. Page 21, "Detection of anti-hIDUA antibodies in plasma". Give volumes of HM-IDUA solution, blocking buffer and antibody solution. Describe method of calculation of data. In Fig. 5f "absorbance" is given, but the concentration of antibodies should be indicated.
12. Figure 1C, legend. Fucose should be α 1,3-linked, as the sample derives from insects. If the MS method could not clearly distinguish between α 1,3- and α 1,6-linkages, omit linkage information for fucose.

Reviewer #4

(Remarks to the Author)

"N-glycan-modified α -L-iduronidase produced by transgenic silkworms ameliorates symptoms in a Japanese macaque with mucopolysaccharidosis I"

COMMSMED-24-0431

This study describes the production of enzyme replacement therapy (ERT) for treatment of mucopolysaccharidosis type I (MPS I) using transgenic silkworm technology and the results of ERT in a previously unpublished model of MPS I in Japanese macaques. The model of MPS I meets diagnostic criteria used in human patients: demonstration of a homozygous

missense mutation in α -IUDA, similar GAG urine profiles, and physical features of abnormal facies, musculoskeletal abnormalities, and cardiac abnormalities. The authors create and test three forms of transgenic human α -L-iduronidase enzymes (hIUDA) for efficacy in vitro using fibroblast lines from an MPS I macaque and in vivo in a single MPS I macaque. The three forms of enzymes, high mannose containing IUDA (HM-IUDA), mannose 6-phosphate modified N-glycan (M6P-IUDA) and sialic acid containing N-glycan (SG-IUDA) are created based on prior evidence of the effects of specific glycosylation types on tissue uptake and efficacy. MPS I fibroblast IUDA enzyme activity is partially restored with M6P-IUDA treatment and SG-IUDA; HM-IUDA is shows essentially no effect. The authors then test the effects of the three forms of hIUDA sequentially in an MPS I macaque with dosing regimens modeled on the commercially available ERT for MPS I (laronidase). Effect of treatment is measured by reduction of urinary glycosaminoglycans and improvement in clinical signs. Anti-hIUDA antibodies are demonstrated.

The study design is innovative and combines the use of transgenic silkworms for potentially larger scale protein production with glycosylation systems to create modified proteins for improved uptake in specific tissues, and the use of a newly discovered nonhuman primate model. However, neither arm of the project is fully developed

1. Since this is a novel model in NHP, more characterization of the model is needed to strengthen the study and utility of the model.

a. Genotyping. The authors apparently only looked for homozygous mutations in a single gene. There is significant overlap in MPS and other LDS phenotypes; excluding other genetic mutations is an important part of determining the significance of the identified mutation in IDUA (MPS II, MPS VII, MPS VI, mucopolisidoses II and III, etc.)

b. Additional discussion of the comparison of the human phenotype and the affected macaques is needed and should be supported with more detailed and current literature. For example, there are excellent references available in this recent paper describing MPS I in a dog.

Mansour TA, Woolard KD, Vernau KL, Ancona DM, Thomasy SM, Sebbag L, Moore BA, Knipe MF, Seada HA, Cowan TM, Aguilar M, Titus Brown C, Bannasch DL. Whole genome sequencing for mutation discovery in a single case of lysosomal storage disease (MPS type 1) in the dog. *Sci Rep.* 2020 Apr 16;10(1):6558. doi: 10.1038/s41598-020-63451-4. PMID: 32300136; PMCID: PMC7162951.

c. In particular, given that corneal opacity is an extremely common feature in MPS I patients, ophthalmological findings should be addressed.

d. The evidence presented by the authors does not adequately preclude possible CNS involvement in their model. Evaluation of mental status and cognition in NHP differs from methods used in humans. It may not be possible to determine if changes in activity are related to musculoskeletal lesions, the CNS, or both. Variations in cognition in NHP that would correlate with learning or intellectual disabilities in humans are difficult to assess in NHP and were not considered. Similarly, methodologies such as in vivo imaging for CNS changes (e.g. hydrocephalus) and gross and microscopic postmortem examination would be necessary to determine the extent of CNS involvement.

e. There is overlap of phenotypes in humans between moderate MPS I (Hurler-Scheie syndrome) and mild MPS I (Scheie syndrome); some consider these as a single combined 'attenuated' MPS I phenotype. This issue should be addressed. Clarke LA. Mucopolysaccharidosis Type I. 2002 Oct 31 [updated 2024 Apr 11]. In: Adam MP, Feldman J, Mirzaa GM, Pagon RA, Wallace SE, Bean LJH, Gripp KW, Amemiya A, editors. *GeneReviews*® [Internet]. Seattle (WA): University of Washington, Seattle; 1993–2024. PMID: 20301341.

f. Information regarding gross and microscopic postmortem findings on the first MPS I animal would be an important addition, if available.

2. The evidence for effects of ERT in the MPS I female is somewhat weak. The most convincing evidence is reduction in urinary GAG. (Note that the significance of Figure 5.e. data is not clear.) Changes in skin elasticity is not presented as a quantifiable attribute so it is difficult to evaluate it as an objective measure of improvement. Similarly, the activity data does not appear compelling as presented. The changes in appearance of gingiva following ERT could potentially be due to advancing age of the animal rather than a treatment effect. Given the small number of affected animals (n=2), it is not possible to determine.

3. Has this IUDA mutation been reported in humans?

Soares LDF, Villalba Silva GC, Kubaski F, Giugliani R, Matte U. MPSBase: Comprehensive repository of differentially expressed genes for mucopolysaccharidoses. *Mol Genet Metab.* 2021 Aug;133(4):372-377. doi: 10.1016/j.ymgme.2021.06.004. Epub 2021 Jun 15. PMID: 34147352.

Borges P, Pasqualim G, Matte U. Which Is the Best In Silico Program for the Missense Variations in IDUA Gene? A Comparison of 33 Programs Plus a Conservation Score and Evaluation of 586 Missense Variants. *Front Mol Biosci.* 2021 Oct 21;8:752797. doi: 10.3389/fmolb.2021.752797. PMID: 34746235; PMCID: PMC8566697.

4. Both aspects of this study (enzyme production and testing and the NHP model) need more development (and might be significantly more compelling if they were separated). Using a mouse or other non-NHP model to test the efficacy of the silkworm transgenic proteins and comparing those to the current ERT with gross and microscopic correlations would be provide strong evidence of potential efficacy. The NHP model is highly valuable for further research but needs to be more thoroughly characterized, including gross and histologic evaluation when available, for best use. It is fully recognized that the issues of using NHP genetic models make this an extraordinarily challenging endeavor. Long reproduction intervals (averaging one pregnancy per year or less), low fecundity (single offspring per offspring), long generation times (sexual maturity at 4 to 7 years of age dependent on sex), and autosomal recessive pattern of inheritance are all tremendous barriers.

5. Minor points

a. 'Symptoms' refer to those subjective things experienced and reported by a patient. 'Signs' are objective characteristics of

disease that can be measured. Because animals can't report what they are experiencing directly to an observer, the convention in veterinary medicine (e.g. in NHP) is to use the term 'signs' in referring to disease manifestations.

b. The gene IDUA is referred to alternately as IDUA and Idua.

c. In Figure 5. d, e, f the purple values are mixed diamonds and circles. It isn't clear if this is an oversight or represents a difference between the two data types.

d. Typographic errors in Supporting information. Legends of Tables S1 and S2: macaques rather than 'macacues'.

Version 1:

Reviewer comments:

Reviewer #1

(Remarks to the Author)

I consider that all my suggestions were attended, I just suggest to include the following aspects:

1. I'm still of the opinion that the recovery of 46% of M6P-IDUA as opposed to WT is not entirely correct, since the SD is so large in M6P-IDUA. I would minimise the result observation in this context. (Results of "Effect of enzyme replacement with hIDUAs on MPS I macaque-derived fibroblasts" related to Figure S2a)
2. In Figure 2b, the "small signal" in the HM-IDUA is not visible, perhaps you can improve the resolution of the image, or perhaps include a quantification graph of the observed signal.
3. Regarding the response about enzyme uptake through mannose receptors, I have to say that your observation is partially true because since mannose receptors are less expressed in fibroblasts, it has been shown that other types of recombinant enzymes with mannose glycosylations have been taken up by human skin fibroblasts through mannose receptors. So you can include some of these reports on yeast expression systems and recombinant proteins.
4. Regarding the response about the relevance of silkworm as an expression system and scaling up of the production process, I suggest that this observation be taken into account: You have to mention that although silkworm could be useful as an expression system of recombinant proteins, the expressed proteins could show instability which may be caused by differences in glycosylation patterns as mentioned in the reference article [15], "...we revealed that the TNFR2-Fc fusion protein expressed by silkworm is profitable but thermally unstable..." which is an important consideration to take into account for scaling up production and industrialisation.

Reviewer #2

(Remarks to the Author)

The authors responded to the majority of the reviewers' requests, which provided clarification on some of the minor or major comments. Nevertheless, it is my opinion that the N-glycan profile is required for both neoglyco-hIDUA, given that the properties of this enzyme are based on transglycosylation. It is essential to demonstrate that N-glycans have undergone modification and to ascertain the extent of these modifications. For example, in Figures 2, 3, and 6, the authors assert that four of the six N-glycosylation sites were perfectly modified in the enzymes with identical N-glycals. Nevertheless, the results of the SDS-PAGE and lectin blot analyses clearly demonstrated that this assertion was inaccurate. Additionally, the in vivo experiments were preliminary in nature, as only a single animal was included, which limited the ability to draw definitive conclusions about the impact of these recombinant enzymes. For example, the results show that in vivo there is not difference between the HM-IDUA compared to the transglycosylated enzymes

Reviewer #3

(Remarks to the Author)

The authors addressed all my points to my complete satisfaction. The reviewer also accepts, that technical limitations did not allow answering all questions raised by the comments of every reviewer, and looks forward to subsequent publications about this highly interesting topic.

Reviewer #4

(Remarks to the Author)

This revised manuscript describes the production of enzyme replacement therapy (ERT) for treatment of mucopolysaccharidosis type I (MPS I) using transgenic silkworm technology and the results of ERT in a previously unpublished model of MPS I in Japanese macaques. The study design is innovative and combines the use of transgenic silkworms for potentially larger scale protein production, glycosylation systems to create modified proteins for improved uptake in specific tissues, and the use of a newly discovered nonhuman primate model. The authors have adequately addressed my major concerns by inclusion of histopathologic evaluation of affected macaques and other revisions in response to all reviewers. I recommend acceptance.

Version 2:

Reviewer comments:

Reviewer #1

(Remarks to the Author)

The authors present a manuscript describing the production of recombinant IDUA obtained from transgenic silkworms, and the transglycosylation method to add Mannose6phosphate glycans and sialic acid to the purified protein. They also show the effect of this modified IDUA in MPS1 macaques as an enzyme replacement therapy.

This article is novel and could be an interesting topic for the scientific community, as it shows a new alternative to obtain N-glycosylated functional protein using non-conventional expression system. However, this host cannot add proper protein glycosylation naturally. Therefore they use the transglycosylation method to add modified glycans in the recombinant protein.

Thus although the protein does not show internalization in the in vitro assays in fibroblasts, it does show some activity in the ERT assay, it does not reach all the desired tissues, which is a good start to demonstrate the effectiveness of the methodology proposed. However, further scaling up is needed to establish production yields and efficiency throughout the production process, including transglycosylation, compared to other expression systems.

Finally, this is a good work that shows another possibility for accessing functional recombinant proteins useful for the treatment of diseases that require enzyme replacement as a therapeutic alternative

Reviewer #2

(Remarks to the Author)

The authors included some important modifications in the revised version. It is clear that they tried to perform the glycan analysis but they could not obtain the protein for those analysis. I'm still consider that those analysis will enrich the manuscript and better support the results. Nevertheless, I consider that the manuscript offer new insights in the field of N-glycosylated enzymes for ERT, which could be later confirm with further preclinical studies. The description of the new animal model for MPS I is also a significant contribution to the field of LSD.

Author Response to Reviewers

We carefully revised our manuscript according to the suggestions of reviewers. We made a note in **blue highlight** where we added to the revised manuscript. Our point-by-point responses to the reviewer comments are as follows.

Reviewer#1:

This manuscript is a well written and describing paper which is focused in assessed a new and low cost expression system to obtain a human recombinant IDUA, with appropriate glycosylation obtained through a transgenic silkworms and a transglycosylation method. Likewise, once the rhIDUA was purified and characterized, they evaluated the therapeutic effect in a non-human primate.

This work has novelty, involving the use of a not commonly used expression system for the production of recombinant proteins and a new method to obtain the desired glycosylation pattern. And the use of MPSI-like macaque to evaluate the effect of the enzyme it's very interesting.

However, in the discussion, the authors assure that the use of the transglycosylation method and silkworm as an expression system allows to obtain a higher homogeneity in the glycosylation pattern. This is only a partial truth, since the immunoblot and SDS-PAGE show a reduction in the smeared band in the elution lane, however is still smeared which indicate the presence of different types of glycosylation in the protein. So maybe the affirmation should be improve the heterogeneity, but not produce homogeneous glycosylation proteins.

(Response) Thanks for pointing this out. We have rephrased “homogeneous” with “functional” in Discussion section.

In Figure S2, where the authors show the effect of IDUA treatment in MPSI macaque cells, it is not entirely clear the significant difference between untreated and treated cells. The bar error in M6P-IDUA untreated is high, so in the end the data obtained is almost equal to the treated cells. Similar results are observed in HM-IDUA and SG-IDUA between treated and non-treated cells. Authors present results of duplicate and triplicate data in the same analysis, which is questionable.

(Response) Thank you for your comment. We excluded the comparison of IDUA activity between HM-IDUA and SG-IDUA-treated cells and presented qualitative data in Results section.

I'm wondering about the results shown in the enzyme delivery images because it has been reported that recombinant proteins can be internalized through mannose receptors, so at least some of the administered HM protein should be taken up. The image of SG-IDUA shows a blue background suggesting overexposure in the microscope. There seems to be no homogeneity between the images.

(Response) Thank you for your comment. Mannose receptors are highly expressed in macrophages and rarely in mammalian skin fibroblasts. Therefore, we believe that HM-IDUA could hardly be incorporated into macaque fibroblasts. Because of the high background for blue fluorescence, we have replaced the images in Supplementary Figure 2b.

So, if the HM-IDUA did not show internalization in fibroblasts, why the authors use this protein in the macaque? what was the purpose of that? Why this protein works in vivo and not in vitro?

(Response) Thank you for your comment. As mentioned above, the mannose receptor is rarely expressed in mammalian skin fibroblasts. On the other hand, We administered HM-IDUA to macaques with the expectation that it would be taken up by cells of the reticuloendothelial system including macrophages that express mannose receptors.

So, in terms of productivity, if this is a promiscuous protein for further ERT to MPSI, is this expression system a good platform to scale up the production of the recombinant enzyme? is the transglycosylation process an efficient method to improve the glycosylation pattern of the protein?

(Response) Thank you for your comment. The production of recombinant enzymes using transgenic silkworms and transglycosylation with ENGases are suitable for scale-up and industrialization. The transglycosylation process is considered an efficient method to introduce functional *N*-glycans into the enzyme. We added them to Discussion section in the revised manuscript.

In terms of macaque evaluation, the use of only one animal (understanding that having an animal model of these diseases is complicated, and bearing in mind that this is a spontaneous model) is a too preliminary data to ensure that this is an effective therapy. So how do we know that if we use the same animal to evaluate three similar proteins, the therapeutic result of the second and/or third protein used is not due to the tolerance developed by the animal because it has previously been injected with other similar proteins, and not inherent to the therapeutic being evaluated?

(Response) Thank you for your comment. To evaluate three enzymes using the same individuals, we allowed a 64-d and 173-d rest period, respectively, between courses. Since total GAG in urine was

increased again after the rest period (Fig. 5d), we believe that the therapeutic effect of the second and third proteins could be evaluated. On the other hand, urinary Hep/HS levels were low at the beginning of the third course (d378, Fig. 5e). Dermatan sulfate (DS) is mainly distributed in the skin, blood vessels, aortic valve, tendons, and lungs. We hypothesize that HS is more readily degraded by M6P-IDUA due to the higher distribution in tissues where HS, but not DS, is accumulated. We determined CS/DS levels and found that CS/DS had increased at d378 and DS had decreased at d434. Therefore, as pointed out, the ERT effect of the third course could be partially due to the tolerance of M6P-IDUA. We added them to Supplementary Figure 3a, Results, and Discussion sections in the revised manuscript.

In the figure 5e Why the urinary GAGs in the third administration start in almost zero and maintain in this low level? why the behaviour is different to the first and second administration where the initial point is higher than the post evaluating days? Between first and second protein the accumulation is recovered, but between second and third maintain low? there is no discussion about that in the manuscript.

(Response) Thank you for your comment. In the first course, the administration of HM-IDUA resulted in a partial reduction of accumulated GAGs, and GAGs were accumulated again before the second course. In the second course, M6P-IDUA was widely distributed, and accumulated GAGs were highly decreased, indicating that urinary Hep/HS was almost absent at the start of the third course. On the other hand, urinary total GAGs and CS/DS were again increased. We added them to Discussion section in the revised manuscript.

Reviewer#2:

In this manuscript, the authors present a novel strategy for producing N-glycosylated recombinant IDUA. Currently, most of the lysosomal enzymes for ERT, including the one approved for MPS I, are produced in mammalian cells, resulting in high prices and low yields. Therefore, it is crucial to develop novel strategies for the production of ERTs at lower prices, which may increase access to therapies. In this context, this manuscript represents a valuable contribution to this field. The description of the novel animal model is also an important contribution to the MPS and LSD fields. Nevertheless, there are some limitations to the impact of the results, and in some cases, the conclusions are not fully supported by the results.

1. Figure 2 illustrates that the treatment with Endo D and Endo M produces a broader band than that of HM-IDUA. This suggests the presence of a non-homogeneous protein, which contrasts with the authors' assertion that this approach may produce homogeneous N-glycans in comparison to other platforms. It is essential to quantify and discuss the homogeneity of the N-glycans present in M6P-

and SG-IDUA. Furthermore, it is necessary to consider the potential implications of this non-homogenous proteins on efficacy, safety, and productivity.

(Response) Thanks for pointing this out. We have rephrased “homogeneous” with “functional” in Discussion section. The observation of a broader band suggests that it is difficult to transglycosylate all six *N*-glycans attached to recombinant IDUA, or to purify only transglycosylated molecules, using current technology. For M6P-IDUA, the yield of purification using Phos-tag was low and the *N*-glycan structure could not be determined. For SG-IDUA, the purification method specific for SG-type *N*-glycans has not been established, and there may be contaminated GlcNAc-IDUA. We believe that further studies using lysosomal proteins with only one *N*-glycan, such as GM2A(GM2 activator protein), as models are needed to examine their potential impact on efficacy, safety, and productivity. We added them to Discussion section in the revised manuscript.

2. It is recommended that the kinetic evaluation of the three recombinant enzymes produced in the manuscript be conducted.

(Response) Thank you for your comment. Pharmacokinetic evaluation using macaques could not be performed due to the need to collect blood samples many times in a short period. Further studies should include pharmacokinetic analysis in mice models. We added them to Discussion section in the revised manuscript.

3. The absence of *N*-glycan profiles for both 6P- and SG-IDUA precludes the drawing of certain conclusions. It is recommended that these profiles be included. With regard to SG-IDUA, the authors posit that cell uptake is mediated by the presence of sialic acid. However, they do not rule out the possibility of mannose receptor-mediated cell uptake, given that it has been observed in high mannose content lysosomal enzymes. The authors must exclude the possibility that the presence of sialic acid increases the half-life of the protein, as this has been widely reported and may facilitate uptake through mannose receptors.

(Response) Thank you for your comment. *N*-glycan profiles for M6P- and SG-IDUA could not be obtained due to issues including purification method and yield as mentioned in comment 1. As pointed out, the high mannose-type *N*-glycans could be incorporated into the reticuloendothelial system via mannose receptors. We consider the HM-IDUA administration in the first course to be the control experiment for the third course of SG-IDUA administration and conclude that a therapeutic effect in the third course of administration is dependent on SG-type glycans. However, the pharmacokinetics of SG-IDUA may have been improved, and we have included these possibilities in Discussion section in the revised manuscript.

4. The authors noted that SG-IDA exhibited reduced activity relative to M6P-IDUA. However, no

statistically significant difference was observed between the two proteins in terms of their activity.

(Response) Thank you for your comment. We excluded the comparison of IDUA activity between M6P- and SG-IDUA treated cells, and presented qualitative data in Results section.

5. If laronidase is administered on a weekly basis, it would be beneficial to understand the rationale behind the administration of the new enzymes every other week.

(Response) Thank you for your comments. Since laronidase is administered to patients with MPS I on a weekly basis, we agree that it should be administered to macaques on a weekly basis as well. Unfortunately, macaques are highly sensitive to general anesthesia and cannot be administered weekly with anesthesia. We added them to Results section in the revised manuscript.

6. The in vivo evaluation was conducted on a single animal that received the three treatments over a period of approximately 15 months. As observed in MPS I patients, it is crucial to consider the natural history of the disease when analyzing these results. Therefore, it is recommended to include the follow-up of a non-treated MPS I animal of the same age and during the same evaluation period. This would permit a more comprehensive analysis of the potential impact of the therapy on GAG levels, skin elasticity, exposure of canines, or daily activity, which are the endpoints highlighted by the authors.

(Response) Thank you for your comment. Since the evaluation period was approximately 15 months, we have investigated a non-treated MPS I macaque (5-year-old, #3 in Fig. 4g) at the endpoint. The urinary GAG levels were increased, and we added them to Fig. 4i. Resistance to puncture suggesting decreased skin elasticity and gingival hyperplasia were observed as in other MPS I macaques. It was difficult to compare the daily activities with that of the enzyme-treated individual because of the considerable differences among individuals.

7. Figure 5C. The authors indicate that there is an increase in activity following the infusion of M6P- and SG-IDUA. However, there is a lack of data regarding HM-IDUA. Additionally, there is an activity increase during the resting period between M6P- and SG-IDUA infusions. This increase merits further discussion.

(Response) Thank you for pointing this out. Unfortunately, we have not analyzed the activity data during the first course (HM-IDUA administration). The increase in activity during the rest period between M6P- and SG-IDUA administration is assumed to be due to the continuous decrease of GAG accumulation in peripheral tissues by M6P-IDUA because the decrease of urinary HS was maintained during the rest period.

8. The authors discuss about the impact on skin elasticity as well as the lack of impact in bone and cardiac disease. Nevertheless, this results are not presented in the manuscript.

(Response) Thank you for your comment. We have added a more concrete description of the change in skin elasticity, with increased resistance to puncture. Because these changes are not presented as a quantifiable attribute, we should examine them in a more quantitative method in future studies. Pathological studies in the mitral valve showed deposits of foamy macrophages characteristic of MPS. We added them to Fig 4j in the revised manuscript. Dr. Akihiro Hirata, who conducted the pathological study, was listed as a co-author.

9. The evidence from in vivo studies is inconclusive regarding the superiority of M6P- or SG-IDUA compared to HM-IDUA. It is uncertain whether M6P- or SG-IDUA will be cost-effective, given the need for additional modification and purification steps, which would increase the cost of the therapy. In this manuscript, the authors present a novel strategy for producing N-glycosylated recombinant IDUA. Currently, most of the lysosomal enzymes for ERT, including the one approved for MPS I, are produced in mammalian cells, resulting in high prices and low yields. Therefore, it is crucial to develop novel strategies for the production of ERTs at lower prices, which may increase access to therapies. In this context, this manuscript represents a valuable contribution to this field. The description of the novel animal model is also an important contribution to the MPS and LSD fields. Nevertheless, there are some limitations to the impact of the results, and in some cases, the conclusions are not fully

(Response) Thank you for your comments. For the superiority of M6P- or SG-IDUA compared to HM-IDUA, further experiments will be required to evaluate the dependence of each on M6P- or SG-type glycans, respectively. For example, it is desirable to perform experiments in which M6P compounds or SG-type glycopeptides are administered simultaneously to another MPS I macaque individual as a competitive inhibitor.

Although the recombinant enzyme production platform using transgenic silkworms is superior in price and yield compared to production in mammalian cells, as pointed out, additional modification and purification steps are required. The yield of purification using Phos-tag was low for M6P-IDUA, and the purification method specific for SG-type N-glycans has not been established for SG-IDUA. It is necessary to address issues related to scale-up and cost performance improvement in the future. We added them to Discussion section in the revised manuscript.

Reviewer#3:

Lysosomal storage disorders are well-known targets for enzyme replacement therapy. In the presented study mucopolysaccharidosis type I, caused by defects in the L-iduronidase (IDUA) gene, is treated with a novel recombinant and glycoengineered IDUA. The authors used transgenic silkworms for the production of high protein amounts and improved its ability for enzyme replacement therapy by chemoenzymatic modifications of IDUA N-glycans. These glycoproteins were applied to enzyme

replacement therapy of mucopolysaccharidosis in an ape model. In summary, it is a comprehensive work, joining glycobiochemistry and glycoengineering with successful use in a promising therapeutic method, suited for medical applications in the future.

There is one major point which should be addressed in a revised version of the manuscript. The authors showed quantitative N-glycan analysis of original recombinant IDUA and of pretreated GlcNAc-IDUA. However, a detailed N-glycan analysis of the completely modified glycoproteins, M6P-IDUA and SG-IDUA, is missed. This is of particular importance, as these proteins are administered in enzyme replacement therapy, and its structure and composition should be known as exact as possible. Phos-tag affinity chromatography or lectin blotting are possible indicators of successful modification of N-glycans. Nevertheless, the amounts of N-glycans carrying mannose 6-phosphate and sialic acid, respectively, which might be influenced by incomplete modification or subsequent degradation, could not be drawn from the experiments.

(Response) Thanks for pointing this out. We have rephrased “homogeneous” with “functional” in Discussion section. The observation of a broader band suggests that it is difficult to transglycosylate all six *N*-glycans attached to recombinant IDUA, or to purify only transglycosylated molecules, using current technology. For M6P-IDUA, the yield of purification using Phos-tag was low and the N-glycan structure could not be determined. For SG-IDUA, the purification method specific for SG-type *N*-glycans has not been established, and there may be contaminated GlcNAc-IDUA. We believe that further studies using lysosomal proteins with only one *N*-glycan, such as GM2A (GM2 activator protein), as models are needed to examine the influences of incomplete modification or subsequent degradation. We added them to Discussion section in the revised manuscript.

Further points:

1. Page 6, chapter “Establishment of IDUA transgenic silkworm and purification of recombinant hIDUA from cocoons” and following figures. Description of N-glycan structures: Give a complete list of abbreviations used for N-glycan structures once in the manuscript. In addition, omit “-PA” from the structures. Use “ α 1,3” and “ α 2,6” without hyphen. Finally, analysis of amino acid sequence is not described in the Methods part.

(Response) Thank you for your comments. We have described a complete list of abbreviations used for *N*-glycan structures without “-PA” in the legend of Fig1. We have corrected “ α -1,3” and “ α -2,6” as “ α 1,3” and “ α 2,6”. We have described the analysis of amino acid sequence in Method section (Site-specific *N*-glycosylation analysis).

2. Page 7. Avoid “as indicated by the Coomassie brilliant blue (CBB) staining”. The method is SDS-

PAGE; CBB is only a part of it, which must not be indicated in the main text when described in the methods. Please check the whole manuscript for respective modifications.

(Response) Thank you for pointing this out. We modified the whole manuscript for respective points.

3. Page 6/7, “Modification of N-glycans...”. Give masses of IDUA used for starting the experiments and their mass yields after the final purification step. This is important to estimate the scale necessary to reveal sufficient amounts of glycoproteins for enzyme replacement therapy.

(Response) Thank you for your comments. We have noted the masses of M6P- and SG-IDUA for starting the experiments and their mass yields after the final purification step in Result section.

4. Methods, general. Intention of the movement of a part of methods to supporting information is unclear for the reviewer. All methods should be joined in one chapter. Furthermore, re-check all chapters of the Methods part for exact description of the methods. Some necessary corrections are listed below, but the reviewer cannot not guarantee to get all gaps. It must be ensured, that all experiments can be completely repeated by other scientists.

(Response) Thank you for your comments. We described a part of the methods in Supplementary information in the original manuscript due to the word limit. We moved them to Methods section in the revised manuscript.

5. Page 15, “After washing, the bound proteins were eluted by increasing the NaPB concentration.” is wrong. The proteins were eluted by decreasing concentrations of ammonium sulfate.

(Response) Thank you for your comments. We have rephrased “increasing the NaPB concentration” with “decreasing concentrations of ammonium sulfate” in the revised manuscript.

6. Page 15, “CBB and silver staining”. Change to “SDS-PAGE analysis” (see 2.). Furthermore, give a reference for CBB staining; there are hundreds of different protocols published.

(Response) Thank you for your comments. We have rephrased “CBB and silver staining” with “SDS-PAGE analysis” in the revised manuscript. We have added detailed reagents and references for CBB staining.

7. Page 16, “Enzyme assays”. Describe the assays in more detail. Ref. 48 is not sufficient here. The activity was measured for purified enzymes and in cell lysates, which requires different protocols.

(Response) Thank you for your comments. The assays are described in more detail in Method section. Purified enzyme and cell lysate are assayed using the same protocol, including dilution with ultrapure water if needed.

8. Page 16, “ENGase treatment”. Use “GlcNAc” instead of GN for N-acetylglucosamine. Define “MP”. Give the exact structure for the α 2,6-sialylglycopeptide. Is it the same structure as shown in Fig. 3? Finally, describe purification method (“was applied to a Hi Trap Butyl FF column”) in detail, or refer to the method described in page 15, if it is identical.

(Response) Thank you for your comments. We have rephrased GN to GlcNAc and MP to methoxyphenyl. We have noted the CAS No. of α 2,6-sialylglycopeptide because the exact structure is a very long notation. We have described that the purification method is identical on page 15.

9. Page 17, “Blotting with...”. Give blocking buffer and its incubation time. Describe “visualized using an anti-biotin HRP-linked antibody” in more detail (exact antibody, concentration, incubation time).

(Response) Thank you for your comments. We have described the composition of the blocking buffer and the incubation time. Note that we did not perform blocking for Phos-tag according to the manufacturer's protocol. We have also described the dilution and incubation times of the antibodies.

10. Page 17, “Site-specific N-glycosylation analysis”. “...desalted using a PD- MiniTrap G-25”; give elution buffer. “0.1% formic acid”; indicate w/v or v/v. „Peptides were analyzed using a TripleTOF 6600 mass spectrometer (Sciex)”; describe method in detail or give a reference.

(Response) Thank you for your comments. The elution buffer for desalting is water. We noted that the formic acid concentration is % (v/v). We added a citation for analysis of the glycopeptide and described the method in detail for confirmation of the amino acid sequence in the Method section.

11. Page 21, “Detection of anti-hIDUA antibodies in plasma”. Give volumes of HM-IDUA solution, blocking buffer and antibody solution. Describe method of calculation of data. In Fig. 5f “absorbance” is given, but the concentration of antibodies should be indicated.

(Response) Thank you for your comments. We have described the volumes of each solution. Because a standard curve cannot be plotted for this method, it is difficult to calculate the concentration of antibodies, and absorbance is used to qualitatively judge positive/negative results. All samples were measured at the same time in this experiment. We have replaced the graph in Figure 5f to add the cut-off value (mean + 2 SD of the wild-type).

12. Figure 1C, legend. Fucose should be α 1,3-linked, as the sample derives from insects. If the MS method could not clearly distinguish between α 1,3- and α 1,6-linkages, omit linkage information for fucose.

(Response) Thank you for your comments. Although the MS method could not clearly distinguish between α 1,3- and α 1,6-linkages, the fucose-containing *N*-glycans were detected as glycopeptides in the PNGase F-digested sample, suggesting that they are probably α 1,3-linked. We have omitted

the linkage information for fucose in the revised manuscript.

Reviewer#4:

"N-glycan-modified α -L-iduronidase produced by transgenic silkworms ameliorates symptoms in a Japanese macaque with mucopolysaccharidosis I"

COMMSMED-24-0431

This study describes the production of enzyme replacement therapy (ERT) for treatment of mucopolysaccharidosis type I (MPS I) using transgenic silkworm technology and the results of ERT in a previously unpublished model of MPS I in Japanese macaques. The model of MPS I meets diagnostic criteria used in human patients: demonstration of a homozygous missense mutation in *–IUDA*, similar GAG urine profiles, and physical features of abnormal facies, musculoskeletal abnormalities, and cardiac abnormalities. The authors create and test three forms of transgenic human α -L-iduronidase enzymes (hIUDA) for efficacy in vitro using fibroblast lines from an MPS I macaque and in vivo in a single MPS I macaque. The three forms of enzymes, high mannose containing IUDA (HM-IUDA), mannose 6-phosphate modified N-glycan (M6P-IUDA) and sialic acid containing N-glycan (SG-IUDA) are created based on prior evidence of the effects of specific glycosylation types on tissue uptake and efficacy. MPS I fibroblast IUDA enzyme activity is partially restored with M6P-IUDA treatment and SG-IUDA; HM-IUDA is shows essentially no effect. The authors then test the effects of the three forms of hIUDA sequentially in an MPS I macaque with dosing regimens modeled on the commercially available ERT for MPS I (laronidase). Effect of treatment is measured by reduction of urinary glycosaminoglycans and improvement in clinical signs. Anti-hIUDA antibodies are demonstrated.

The study design is innovative and combines the use of transgenic silkworms for potentially larger scale protein production with glycosylation systems to create modified proteins for improved uptake in specific tissues, and the use of a newly discovered nonhuman primate model. However, neither arm of the project is fully developed

1. Since this is a novel model in NHP, more characterization of the model is needed to strengthen the study and utility of the model.

a. Genotyping. The authors apparently only looked for homozygous mutations in a single gene. There is significant overlap in MPS and other LDS phenotypes; excluding other genetic mutations is an important part of determining the significance of the identified mutation in *IDUA* (MPS II, MPS VII, MPS VI, mucopolysaccharidosis II and III, etc.)

(Response) Thank you for your comments. We have confirmed LSD-related genes with overlapping phenotypes (Supplementary Table 1). Among 21 genes, we found only c.786C>A missense mutation in *IDUA*, which may be a de novo mutation or recessive inheritance in the macaque. There were

heterozygous (c.786C/A) and a few homozygous (c.786A/A) individuals only in the group which originated from Wakasa and not in the other groups. We added them to Results section in the revised manuscript.

b. Additional discussion of the comparison of the human phenotype and the affected macaques is needed and should be supported with more detailed and current literature. For example, there are excellent references available in this recent paper describing MPS I in a dog.

Mansour TA, Woolard KD, Vernau KL, Ancona DM, Thomasy SM, Sebbag L, Moore BA, Knipe MF, Seada HA, Cowan TM, Aguilar M, Titus Brown C, Bannasch DL. Whole genome sequencing for mutation discovery in a single case of lysosomal storage disease (MPS type 1) in the dog. *Sci Rep*. 2020 Apr 16;10(1):6558. doi: 10.1038/s41598-020-63451-4. PMID: 32300136; PMCID: PMC7162951.

(Response) Thank you for your comment. We added the examination for corneal opacity noted in comment 1c and the CNS signs noted in comment 1d to the results section, and added a discussion comparing the human phenotype and the affected macaques.

c. In particular, given that corneal opacity is an extremely common feature in MPS I patients, ophthalmological findings should be addressed.

(Response) Thank you for your comments. Corneal opacity is one of the common features of patients with MPS I. However, there were no findings of corneal opacity in MPS I macaques between the ages of 5 to 8 years. We added them to Results section in the revised manuscript.

d. The evidence presented by the authors does not adequately preclude possible CNS involvement in their model. Evaluation of mental status and cognition in NHP differs from methods used in humans. It may not be possible to determine if changes in activity are related to musculoskeletal lesions, the CNS, or both. Variations in cognition in NHP that would correlate with learning or intellectual disabilities in humans are difficult to assess in NHP and were not considered. Similarly, methodologies such as in vivo imaging for CNS changes (e.g. hydrocephalus) and gross and microscopic postmortem examination would be necessary to determine the extent of CNS involvement.

(Response) Thank you for your comment. We have described morphology of the brain obtained by T1-weighted MRI. There were no signs of hydrocephalus in the MPS I macaque (#1). In the other two macaques (#2 and #3), there was mild expansion of lateral ventricles, but the degree is as that sometimes found in healthy Japanese macaques. Though we did not test their cognitive function systematically, we found no abnormality in taking food pellets from the box outside the cage or in their social relationship with cagemate macaques or caretakers. Also, there were no signs of spinal canal stenosis. On the other hand, there was obvious knee joint contracture. Thus, we suggest the

activity change in the MPS I macaque was mainly due to musculoskeletal disorders and the effects of neurological disorders are minimal, if any.

e. There is overlap of phenotypes in humans between moderate MPS I (Hurler-Scheie syndrome) and mild MPS I (Scheie syndrome); some consider these as a single combined ‘attenuated’ MPS I phenotype. This issue should be addressed.

Clarke LA. Mucopolysaccharidosis Type I. 2002 Oct 31 [updated 2024 Apr 11]. In: Adam MP, Feldman J, Mirzaa GM, Pagon RA, Wallace SE, Bean LJH, Gripp KW, Amemiya A, editors. GeneReviews® [Internet]. Seattle (WA): University of Washington, Seattle; 1993–2024. PMID: 20301341.

(Response) Thank you for your comments. We rewrote “mild type of MPS I (Scheie disease)” to “attenuated type of MPS I (Scheie disease or Hurler/Scheie disease)” in the Results section in the revised manuscript.

f. Information regarding gross and microscopic postmortem findings on the first MPS I animal would be an important addition, if available.

(Response) Thank you for your comments. At necropsy, we observed thickened soft palate, liver nodules, and deformed heart valves in MPS I macaques. Pathological studies in the gingiva and mitral valve showed deposits of foamy macrophages characteristic of MPS. We added them to Fig 4j in the revised manuscript. Dr. Akihiro Hirata, who conducted the pathological study, was listed as a co-author.

2. The evidence for effects of ERT in the MPS I female is somewhat weak. The most convincing evidence is reduction in urinary GAG. (Note that the significance of Figure 5.e. data is not clear.) Changes in skin elasticity is not presented as a quantifiable attribute so it is difficult to evaluate it as an objective measure of improvement. Similarly, the activity data does not appear compelling as presented. The changes in appearance of gingiva following ERT could potentially be due to advancing age of the animal rather than a treatment effect. Given the small number of affected animals (n=2), it is not possible to determine.

(Response) Thank you for your comments. We have added a more concrete description of the change in skin elasticity, with increased resistance to puncture. Because these changes are not presented as a quantifiable attribute, we should examine them in a more quantitative method in future studies. To test whether the change in gingival appearance after ERT was due to treatment effect or advancing age, we have evaluated the ERT macaque before and long-time after ERT and other MPSI macaques. Gingival hyperplasia worsened with age and the change we found at 84 d of ERT was rescue of the hyperplasia. Thus the rescue at 84 d could be ERT effect. We add them to Fig 5b and Results

section in the revised manuscript.

3. Has this IUDA mutation been reported in humans?

Soares LDF, Villalba Silva GC, Kubaski F, Giugliani R, Matte U. MPSBase: Comprehensive repository of differentially expressed genes for mucopolysaccharidoses. *Mol Genet Metab.* 2021 Aug;133(4):372-377. doi: 10.1016/j.ymgme.2021.06.004. Epub 2021 Jun 15. PMID: 34147352.

Borges P, Pasqualim G, Matte U. Which Is the Best In Silico Program for the Missense Variations in IDUA Gene? A Comparison of 33 Programs Plus a Conservation Score and Evaluation of 586 Missense Variants. *Front Mol Biosci.* 2021 Oct 21;8:752797. doi: 10.3389/fmolb.2021.752797. PMID: 34746235; PMCID: PMC8566697.

(Response) Thank you for your comments. Although the H262 residue of IDUA is conserved between humans and Japanese macaques, no mutations in H262 have been reported in patients with MPS I in the above papers and the Human Gene Mutation Database. We predicted the H262Y and H262R variants in humans to be likely pathogenic using the AlphaMissense tool (score, 0.9126 and 0.9349). In addition, the H262 residue in hIDUA has been reported to be positioned near the substrate binding site, it was suggested that mutant IDUA (p. H262Q) has reduced affinity for substrates. We added them to Results section in the revised manuscript.

4. Both aspects of this study (enzyme production and testing and the NHP model) need more development (and might be significantly more compelling if they were separated). Using a mouse or other non-NHP model to test the efficacy of the silkworm transgenic proteins and comparing those to the current ERT with gross and microscopic correlations would be provide strong evidence of potential efficacy. The NHP model is highly valuable for further research but needs to be more thoroughly characterized, including gross and histologic evaluation when available, for best use. It is fully recognized that the issues of using NHP genetic models make this an extraordinarily challenging endeavor. Long reproduction intervals (averaging one pregnancy per year or less), low fecundity (single offspring per offspring), long generation times (sexual maturity at 4 to 7 years of age dependent on sex), and autosomal recessive pattern of inheritance are all tremendous barriers.

(Response) We agree with the comment that it would be useful to test the efficacy and safety of the recombinant enzymes produced by transgenic silkworms using MPS I model mice and wild-type non-human primate models. In this study, we discovered the MPS I macaques which have similar clinical signs as patients with MPS I, and evaluated the efficacy of modified recombinant enzymes. Although the enormous cost, years, and effort required to use the MPS I macaque model, we obtained valuable data. In the future, it will be necessary to combine MPS I model mice, wild-type non-human primate models, and MPSI macaques to evaluate the efficacy and safety of other modalities such as

gene therapy. We added them to Discussion section in the revised manuscript.

5. Minor points

a. ‘Symptoms’ refer to those subjective things experienced and reported by a patient. ‘Signs’ are objective characteristics of disease that can be measured. Because animals can’t report what they are experiencing directly to an observer, the convention in veterinary medicine (e.g. in NHP) is to use the term ‘signs’ in referring to disease manifestations.

(Response) Thanks for pointing this out. We have rephrased “symptom” with “sign” in the revised manuscript.

b. The gene IDUA is referred to alternately as IDUA and *Idua*.

(Response) Thank you for your comments. We have revised the human and macaque genes to *IDUA* and the mouse gene to *Idua* in the revised manuscript.

c. In Figure 5. d, e, f the purple values are mixed diamonds and circles. It isn’t clear if this is an oversight or represents a difference between the two data types.

(Response) Thank you for your comment. The purple diamonds and gray circles indicate the time points measured with SG-IDUA administration and without enzyme, respectively. We have corrected the legend in the revised manuscript.

d. Typographic errors in Supporting information. Legends of Tables S1 and S2: macaques rather than ‘macacues’.

(Response) Thank you for your comments. We have corrected “macacues” to “macaques”.

Author Response to Reviewers

We carefully revised our manuscript according to the suggestions of reviewers. We made a note in **green highlight** where we added to the revised manuscript. Our point-by-point responses to the reviewer comments are as follows.

Reviewer #1:

I consider that all my suggestions were attended, I just suggest to include the following aspects:

1. I'm still of the opinion that the recovery of 46% of M6P-IDUA as opposed to WT is not entirely correct, since the SD is so large in M6P-IDUA. I would minimise the result observation in this context. (Results of "Effect of enzyme replacement with hIDUAs on MPS I macaque-derived fibroblasts" related to Figure S2a)

(Response) Thanks for pointing this out. We have minimized the result observation in Results section.

2. In Figure 2b, the "small signal" in the HM-IDUA is not visible, perhaps you can improve the resolution of the image, or perhaps include a quantification graph of the observed signal.

(Response) Thank you for your comment. We have rephrased "little signal" with "almost no signal" in Results section.

3. Regarding the response about enzyme uptake through mannose receptors, I have to say that your observation is partially true because since mannose receptors are less expressed in fibroblasts, it has been shown that other types of recombinant enzymes with mannose glycosylations have been taken up by human skin fibroblasts through mannose receptors. So you can include some of these reports on yeast expression systems and recombinant proteins.

(Response) Thank you for your comment. We have cited several papers on recombinant enzymes taken up by fibroblasts via the mannose receptor in Discussion section (Shen, J.S. et al. 2016, Rodríguez-López, A. et al. 2016, Pimentel-Vera, L.N. et al. 2024).

4. Regarding the response about the relevance of silkworm as an expression system and scaling up of the production process, I suggest that this observation be taken into account: You have to mention that although silkworm could be useful as an expression system of recombinant proteins, the expressed proteins could show unstability which may be caused by differences in glycosylation patterns as mentioned in the reference article [15], "...we revealed that the TNFR2-

Fc fusion protein expressed by silkworm is profitable but thermally unstable..." which is an important consideration to take into account for scaling up production and industrialisation.

(Response) Thank you for your comment. We mentioned the following. "Transgenic silkworms could be useful as an expression system of recombinant proteins, the expressed proteins could show instability which may be caused by differences in glycosylation patterns¹⁵,"

Reviewer #2

The authors responded to the majority of the reviewers' requests, which provided clarification on some of the minor or major comments. Nevertheless, it is my opinion that the N-glycan profile is required for both neoglyco-hIDUA, given that the properties of this enzyme are based on transglycosylation. It is essential to demonstrate that N-glycans have undergone modification and to ascertain the extent of these modifications. For example, in Figures 2, 3, and 6, the authors assert that four of the six N-glycosylation sites were perfectly modified in the enzymes with identical N-glycals. Nevertheless, the results of the SDS-PAGE and lectin blot analyses clearly demonstrated that this assertion was inaccurate. Additionally, the *in vivo* experiments were preliminary in nature, as only a single animal was included, which limited the ability to draw definitive conclusions about the impact of these recombinant enzymes. For example, the results show that *in vivo* there is not difference between the HM-IDUA compared to the transglycosylated enzymes.

(Response) Thank you for your comment. *N*-glycan profiles for M6P- and SG-IDUA could not be obtained due to issues including purification method and yield. For M6P-IDUA, the yield of purification using Phos-tag was low and the *N*-glycan structure could not be determined. For SG-IDUA, the purification method specific for SG-type *N*-glycans has not been established, and there may be contaminated GlcNAc-IDUA.

In the original manuscript, we described a higher homogeneity in the glycosylation pattern. However, we have rephrased "homogeneous" with "functional" in the revised manuscript. We also mentioned the observation of a broader band suggests that it is difficult to transglycosylate all six *N*-glycans attached to recombinant IDUA, or to purify only transglycosylated molecules, using current technology.

To evaluate three enzymes using the same individuals, we allowed a 64-d and 173-d rest period, respectively, between courses. Since total GAG in urine was increased again after the rest period (Fig. 5d), we believe that the therapeutic effect of the second and third proteins could be evaluated.

On the other hand, urinary Hep/HS levels were low at the beginning of the third course (d378, Fig. 5e). Dermatan sulfate (DS) is mainly distributed in the skin, blood vessels, aortic valve, tendons, and lungs. We hypothesize that HS is more readily degraded by M6P-IDUA due to the higher distribution in tissues where HS, but not DS, is accumulated. We determined CS/DS levels and found that CS/DS had increased at d378 and DS had decreased at d434. Therefore the ERT effect of the third course could be partially due to the tolerance of M6P-IDUA.